evolution, palaeontology, ecology

florivory, fossil plant damage, functional-feeding group, generalist, nectar robbing, pollination

**Authors for correspondence:**
Conrad Labandeira
e-mail: labandec@si.edu
Dong Ren
e-mail: rendong@cnu.edu.cn

# Florivory of Early Cretaceous flowers by functionally diverse insects: implications for early angiosperm pollination

Lifang Xiao[1,2], Conrad Labandeira[1,2,3], David Dilcher[4] and Dong Ren[1]

[1]College of Life Science and Academy for Multidisciplinary Studies, Capital Normal University, Beijing, People's Republic of China
[2]Department of Paleobiology, National Museum of Natural History, Smithsonian Institution, Tenth Street and Constitution Avenue, Washington, DC, USA
[3]Department of Entomology, University of Maryland, College Park, MD, USA
[4]Department of Geology, Indiana University, 1001 Tenth Street, Bloomington, IN, USA

LX, 0000-0002-6940-1473; CL, 0000-0002-4838-5099; DD, 0000-0001-7226-6703; DR, 0000-0001-8660-0901

Florivory (flower consumption) occurs worldwide in modern angiosperms, associated with pollen and nectar consumption. However, florivory remains unrecorded from fossil flowers since their Early Cretaceous appearance. We test hypotheses that earliest angiosperms were pollinated by a diverse insect fauna by evaluating 7858 plants from eight localities of the latest Albian Dakota Formation from midcontinental North America, in which 645 specimens (8.2%) were flowers or inflorescences. Well-preserved specimens were categorized into 32 morphotypes, nine of which displayed 207 instances of damage from 11 insect damage types (DTs) by four functional-feeding groups of hole feeding, margin feeding, surface feeding and piercing-and-sucking. We assessed the same DTs inflicted by known florivores on modern flowers that also are their pollinators, and associated insect mouthpart types causing such damage. The diverse, Dakota florivore–pollinator community showed a local pattern at Braun's Ranch of flower morphotypes 4 and 5 having piercing-and-sucking as dominant and margin feeding as minor interactions, whereas *Dakotanthus cordiformis* at Rose Creek I and II had an opposite pattern. We found no evidence for nectar robbing. These data support the rapid emergence of early angiosperms of florivore and associated pollinator guilds expressed at both the local and regional community levels.

## 1. Introduction

Flowers are the most successful plant reproductive structures ever to evolve on land [1] and angiosperms (flowering plants) presently are the most abundant and diverse clade of vascular plants [2], currently consisting of over 369 000 described species [1]. This lineage probably originated very early during the Cretaceous, with robust molecular phylogenies placing the origins of the clade at 139.35–136 Ma [3]. This timing is consistent with the earliest documented appearance of angiosperm pollen around 136 Ma [4] and earliest known intact flowers at 125 Ma, from the Early Cretaceous of northeastern China [2,5]. The best documented and earliest known bisexual flower, the 'Rose Creek Flower', from the Early Cretaceous Dakota Formation of the United States, examined in this report, is approximately 103 million years (103 Ma) in age [6,7]. Despite the long and sporadic record of fossil flowers and given the abundance and diversity of Dakota Formation flowers, the time is propitious for examination of insect florivory (electronic supplementary material, text S1), which is the consumption of flowers prior to seed coat formation [8], and associated pollination in probably the earliest angiosperm deposit conducive to such an assessment. Such an evaluation could provide a

better understanding of the role that mutualisms and antagonisms of angiosperms, effected by their insect pollinators, had on their joint diversification [9].

Here, three hypotheses are posed that we seek to test in our study of Dakota Formation flowers and inflorescences. The first hypothesis is 'general features of insect florivory and related pollination, as measured by damage patterns on Dakota flowers, are very similar or the same as those made by modern florivores'. The second hypothesis is 'major taxonomic groups of insect florivores and pollinators from the Dakota Formation are very similar or the same as those of today'. The third, more focused, hypothesis is 'nectar robbing was present on Dakota flowers'. By addressing these three hypotheses, we place this study in a broader context of Cretaceous angiosperm pollination and provide a glimpse into early angiosperm florivory and associated pollination about 30 million years after the earliest appearance of angiosperms.

## 2. Material and methods

### (a) The Dakota formation

All compression or impression specimens of flowers and other reproductive material (heretofore termed flower specimens) described in this report were collected from the Dakota Formation of midcontinental United States at eight localities, each bearing a flora of mid-Cretaceous (latest Albian to earliest Cenomanian) age [7], equivalent to approximately 104–97 Ma [10], depending on the locality (electronic supplementary material, figure S1 and table S1). Flower specimens were collected during the 1970s and 1980s by D. Dilcher and colleagues and are stored at the Florida Museum of Natural History, in Gainesville, Florida. The localities are distributed along a younger to older north–northeast to south–southwest transect [11] (electronic supplementary material, figure S1). The localities were deposited along the eastern coast of the North American Mid Cretaceous Seaway that extended from the Gulf of Mexico to the Canadian Arctic [12,13].

Lithological facies within the Dakota Formation are dominated by shales and sandstones of the upper Janssen Clay Member and subjacent claystones of the lower Terra Cotta Clay Member [11,13]. Dakota depositional environments are represented by brackish estuaries, freshwater swamps, low energy channels, floodplain ponds and ox-bow lakes (electronic supplementary material, table S1). These biotic environments consisted of humid forests and woodland [11,13]. The Rose Creek I locality, where most of the flowers were deposited, represented a mangal-like marsh similar to extant communities in southeast Asia [14].

### (b) Dakota formation floras

Dakota Formation floras occur in a northeastern to southeastern trend at Courtland, Minnesota; Pleasant Dale, Nebraska (NE); Rose Creek I and II, NE; Braun's Ranch, Kansas (KS); Linnenberger Brothers' Ranch, KS; Acme Brick Quarry, KS; and Hoisington, KS [11] (electronic supplementary material, figure S1). Known vascular plant species diversity throughout the Dakota localities consists of 134 species. An earlier conservative estimate is 150–200 angiosperm species/morphotypes present in the Dakota Formation, as there is less than 25% species overlap between any two localities [6,7,11,15–18] (electronic supplementary material, table S1). Previous occasional descriptions of flowers, inflorescences, infructescences and fruits are known from several studies [7,19,20], yet affiliations of vegetative taxa with reproductive taxa remain largely unknown, although a few mostly wind-pollinated flowers, are associated with vegetative material that have been described [21]. Except for one locality, localities with higher abundance and

diversity of foliage also appear to have a higher diversity and abundance of floral morphotypes (electronic supplementary material, figure S2 and table S1). (In this report, we use the term, floral, to refer to a flower, and not a bulk flora involving principally foliage.)

The only described and best-known flower from the Dakota Formation is *Dakotanthus cordiformis* [7]. This flower has five, crescent-shaped, nectariferous pads that occur at the base of the gynoecium, each of which is aligned with a sepal. *Dakotanthus*, the most abundant morphotype in our dataset, is a member of the Rosidae 1 clade [7] and apparently very similar to a modern taxon with a lobed nectary disc. Other Dakota flower morphotypes show poor development or apparent absence of nectaries or nectary-like structures. However, leaf taxa occurring in the same localities as the unaffiliated Dakota flowers and infructescences have been assigned to extant families within Austrobaileyales, Chloranthales, Canellales, Magnoliales, Laurales and Rosidae 1 [15], which share a common pattern of fluid rewards for pollinating insects [22]. This pattern consists of: (i) staminoidal appendages (sterile stamens) that produce at their base glandular secretions of nectar-like fluids, mucilage, or 'viscous substances'; (ii) nectariferous glands at the base or tips of fertile stamens; (iii) stigmas that secrete nectar-like substances, usually at their tips; (iv) nectar secreting, parenchymatous tissue on the adaxial surfaces of petals or sepals; and (v) large, substantive glands at the base of stamens that would qualify as true nectaries [22]. From these observations, it is highly likely that Dakota flower morphotypes produced nectar or other secretory, nectar-like fluids that attracted insect florivores and pollinators.

### (c) Identification of insect-damaged flowers and possible culprits

The collection of florivory data is analogous to data for foliage or other vegetative organs and follows the same system of evaluating plant–insect associations [23], extensively used in fossil herbivory studies [24]. This system uses the functional-feeding group (FFG)–damage type (DT) system in which the overarching unit of herbivory is the FFG, examples of which are hole feeding, margin feeding, surface feeding and piercing-and-sucking for Dakota flower damage. Each FFG encompasses several or more DTs, which are the basic units of damage for fossil herbivory studies. A DT may be used in three ways. First, a DT may be used in terms of *DT richness*, referring to the kinds of DTs present; or as *DT occurrences*, as in the individual instances of damage of on a leaf; or as a *formal name*, such as DT405, which is a defined, specific mode of margin feeding damage. Details of photodocumentation and statistical methods are given in the electronic supplementary material, text S2. For Dakota plants, previous assessments of herbivory involved almost entirely mining damage on leaves [25–27]. However, Dakota plants, similar to amber deposits [28,29], provide considerable indirect evidence for flower–insect associations in the fossil record.

### (d) Distinguishing florivores and pollinators

Insect visitors to flowers are of two fundamental groups, florivores and pollinators [30]. Not all florivores are pollinators and not all pollinators are florivores, and the relationships between these two ecological guilds are complex [8]. Florivores typically leave damage on flowers, overwhelmingly on petals [31], often resulting in negative interactions [32]. However, some florivore interactions are neutral or even positive [31,33], as petals occasionally contain nutritive or highly scented tissues designed for consumption by florivores as pollinators [34,35]. Florivory can be a form of predation if plant embryonic tissues are destroyed before the opening of the flower, or if there is the consumption of immature pollen, features that do not appear present in bowl-shaped Dakota flowers, as

the damage is overwhelmingly on inner petal surfaces. Consequently, florivores such as Orthoptera (katydids), Hemiptera (aphids, bugs), Thysanoptera (thrips), Coleoptera (beetles) and Hymenoptera (sawflies, wasps, bees) with mandibulate, stylate or similarly modified mouthparts [36], provide good proxy data for the broad spectrum of pollinator interactions on flowers [37] (table 1). However, a substantial component such as most adult Diptera [51] and Lepidoptera are nondamagers, as they do not leave damage on flowers.

## 3. Results

### (a) Total and insect-damaged flower morphotype abundance and diversity

The eight localities of the Dakota Formation consist of approximately 7858 total plant specimens, which yielded 645 (8.2%) flower specimens that were assessed for insect damage, some of which were photographically documented (electronic supplementary material, text S2). Flower specimens previously were identified to morphotype by Dilcher and Manchester [10,36], and Xiao, but mostly by the latter. The plant specimens were categorized into 32 flower morphotypes, one of which was *Dakotanthus cordiformis* (figure 1; electronic supplementary material, figures S2, S3, S6 and appendix 1). The 32 morphotypes consisted of *Dakotanthus cordiformis* [6,7], 14 flower morphotypes, eight inflorescence–infructescence morphotypes, five reproductive morphotypes, two flower–seed–fruit morphotypes and two Braun Ranch flower morphotypes. Unidentifiable specimens and poor preserved morphotypes, not assigned to one of the 32 morphotypes, were Acme unidentified inflorescence–infructescence, unidentified flower and unidentified stamen, which amounted to one, eight and nine specimens, respectively, attributable to a very limited local sample size or poor preservation. Based on the diversity and abundance of floral morphotypes (electronic supplementary material, figures S2 and S3), our estimate of the log normal fit is 182 species. We also obtained a Fisher's α value of 7.94 (electronic supplementary material, appendix S2).

### (b) General patterns

Insect damage was present on 109 of the 645 examined flower and related specimens, for a specimen-based florivory rate of 17.2%. This damage was represented by four FFGs, 11 DTs [23] and 207 individual DT occurrences (electronic supplementary material, table S2). Some DTs occurred multiple times on the same specimen. One or more DTs on a specimen was present on nine of the 32 flower morphotypes (28.1%) (electronic supplementary material, table S2). The four FFGs present were hole feeding, margin feeding, surface feeding and piercing-and-sucking (see the electronic supplementary material, tables S2, S3 and text S3 for additional DT occurrence details).

### (c) Assessing florivore host specificity by flower morphotype

The distribution of DTs on plant hosts revealed three levels of host specificity [23]. Borrowing from studies of fossil herbivory as an example [24], host specificity is categorized as specialized damage if three or more occurrences of the same DT are present on the same host morphotype or on a very closely related host; damage is of intermediate specificity if the distribution of three

or more occurrences of the same DT are present on more distantly related hosts; and generalized damage if three or more occurrences of the same DT are present on unrelated hosts [23]. For Dakota folivory data, because the phylogenetic relationships among flower morphotypes are unknown, terms expressing host specificity are referenced to the distribution of DTs on the flower species and morphotypes (electronic supplementary material, table S2). The three examples of specialist damage are small hole-feeding DT01 on flower morphotype 4 that hosts 12 of 13 (92.3%) of all occurrences; circular holes between 1 and 5 mm in diameter on *Dakotanthus* that hosts all eight (100%) of occurrences; and notched margin feeding of DT405 along the petal edges on *Dakotanthus*, which hosts 66 of 67 (98.5%) of all occurrences. The single example of damage of intermediate specificity is DT12, evidenced by too few DT distributions across three flower morphotypes. Seven examples of generalized damage are present. They are single, random, piercing-and-sucking damage assigned to DT46 on *Dakotanthus* and flower morphotypes 6, 9 and 10; clustered piercing-and-sucking assigned to DT402 on *Dakotanthus* and flower morphotypes 1, 5 and 8; and DT13, DT29, DT48, DT138 and DT383 that defaults to generalized specificity, each having only one or two occurrences on *Dakotanthus* and flower morphotypes 4, 5 and 7. This pattern of host specificity indicates three examples of specialized damage, one of intermediate specificity damage, and seven of generalized damage (electronic supplementary material, table S2).

### (d) Assessing the geographical distribution of florivory by locality

Of the eight localities examined, flower morphotypes from three localities—Rose Creek I, Rose Creek II and Braun's Ranch—showed evidence of florivory. The combined Rose Creek I and II localities exhibited three flower morphotypes with 117 DT occurrences, whereas the Braun's Ranch locality showed a higher diversity of seven flower morphotypes and 90 DT occurrences (electronic supplementary material, table S2). The number of florivorized to total flower morphotypes at each locality (electronic supplementary material, table S2)—three of 10 (30%) at Rose Creek I and II, and seven of 13 (53.8%) at Braun's Ranch—are distinctly significant subsets of the number of available hosts at each locality. At Rose Creek, *Dakotanthus* overwhelmingly was the dominant flower morphotype present, which displayed a rich spectrum of damage, with three of the four FFGs and seven of the 11 DTs represented (electronic supplementary material, table S2). By comparison, Braun's Ranch showed two florivorized flower morphotypes with a less rich spectrum of FFGs and DTs. Flower morphotype 4 had three of four FFGs and five of 11 DTs present. Similarly, flower morphotype 5 displayed three of four FFGs and six of 11 DTs. These latter flower morphotypes from the Rose Creek and Braun's Ranch localities exhibited a similar distribution of FFGs and DTs.

### (e) Assessing functional-feeding group and damage type on flower morphotypes
#### (i) Hole feeding
Hole feeding on flowers of the Dakota Formation mostly is single, small and circular perforations of the entire petal thickness that are ovate or circular in shape (electronic supplementary material, text S4). Reaction rims are variably developed and occasionally associated with necroses of

**Table 1.** The potential florivory and pollination insect taxa from fossil and modern evidence.

| pollinator clade present by ≈105 Ma | evidence[a] | | | pollinator effectiveness[b] | | | FFGs[c] | | | | | major reward[d] | | encompassing mouthpart classes[e] | general sources[f] |
|---|---|---|---|---|---|---|---|---|---|---|---|---|---|---|---|
| | florivory | phylogeny | fossils | min | inter | max | HF | MF | SF | P&S | ND | pollen | nectar | | |
| **ORTHOPTERA** | | | | | | | | | | | | | | | |
| Gryllacrididae | X | | | X | | | X | X | | | | | X | adult ectognathate | [38] |
| Tettigoniidae | X | X | | X | | X | X | X | | | | | X | (nymphs and adults) | [38] |
| Acrididae | X | X | | X | | | X | X | | | | | X | | [38] |
| **THYSANOPTERA** | | | | | | | | | | | | | | | |
| Melanthripidae | | X | X | | | X | X | X | | X | | X | | mouthcone (nymphs and adults) | [28] |
| Thripidae | X | X | X | | | X | X | X | | X | | X | | | [39] |
| **HEMIPTERA** | | | | | | | | | | | | | | | |
| Aphididae | X | X | X | X | | | X | X | | X | | | X | segmented beak | [40] |
| Lygaeidae | X | X | X | | X | | X | X | | X | | | X | (nymphs and adults) | [41] |
| Miridae | X | X | X | | | X | X | X | | X | | | X | | [42] |
| Pentatomoidea | X | X | X | | | X | X | | | X | | | X | | [43] |
| **COLEOPTERA** | | | | | | | | | | | | | | | |
| Staphylinidae | X | | | X | | | X | | X | | | | X | larval ectognathate | [44] |
| Scarabaeidae | X | | | | | X | X | X | X | | | X | | (larvae), adult ecto-gnathate (adults), | [44] |
| Kateretidae | | | X | | X | | X | X | X | | | X | | rhynchophorate | [44] |
| Tenebrionoidea[g] | X | X | X | | X | | X | X | X | | | X | | (adults) | [44] |
| Chrysomeloidea | X | | | | X | | X | X | X | | | | X | | [44] |
| Curculionoidea | X | X | X | | | X | X | X | X | | | | X | | [44] |
| **HYMENOPTERA** | | | | | | | | | | | | | | | |
| Xyelidae | X | X | X | X | | | | | | | | X | | sericterate (larvae) | [45] |
| Megalodontesidae | | | X | | | | | | | | | X | | adult ectognathate | [45] |
| Pamphiliidae | | X | X | | X | | | | | | | | X | (adults) glossate | [45] |
| Tenthredinoidea | | X | | | | X | | | | | | | X | (adults) maxillo- | [45] |
| Siridae | | X | | | X | | | | | | | | X | labiate (adults) | [45] |
| Formicidae | X | X | | X | | | X | X | | | | | X | | [46] |
| Apoidea | X | X | | | X | X | X | X | | | | | X | | [46] |
| **LEPIDOPTERA** | | | | | | | | | | | | | | | |
| Micropterygidae | | X | X | X | | | X | | | | X | X | | larval ectognathate | [47] |
| Agathiphagidae | | X | | X | | | X | X | | | X | X | | (larvae) sericterate | [47] |
| Heterobathmiidae | | X | | X | | | X | X | | | X | X | | (larvae) adult ecto-gnathate (adults), | [47] |
| Nepticuloidea | | X | X | | X | | | | | | X | | X | siphonate (adults) | [47] |
| Eulepidoptera[h] | X | X | | | | X | X | X | | | | | X | | [47] |
| Yponmeutoidea | | X | | | X | | | | X | | | | X | | [47] |
| Gracillarioidea | | X | X | | X | | | | X | | X | | X | | [47] |
| Tortricidae | | X | | | | X | | | X | | | | X | | [47] |

(*Continued.*)

**Table 1.** (Continued.)

| pollinator clade present by ≈105 Ma | evidence[a] | | | pollinator effectiveness[b] | | | FFGs[c] | | | | | major reward[d] | | encompassing mouthpart classes[e] | general sources[f] |
|---|---|---|---|---|---|---|---|---|---|---|---|---|---|---|---|
| | florivory | phylogeny | fossils | min | inter | max | HF | SF | MF | P&S | ND | pollen | nectar | | |
| **DIPTERA** | | | | | | | | | | | | | | | |
| Nematocera | | | x | | | x | | | | | x | | x | monostylate/distylate | [48] |
| Stratiomyidae | | | | x | | | | | | | x | | x | (adults), distylate/ | [48] |
| Asiloidea | | | x | | | x | | | | | x | | x | tetrastylate (adults) | [48] |
| Rhagionidae | | | | | x | | | | | | x | | x | hexastylate (adults) | [48] |
| Eremoneura | | | | x | | | | | | | x | | x | labellate (adults) | [48] |

[a]Evidence for florivory is documented in the electronic supplementary material, table S3. Myanmar amber is sufficiently close in time to the Dakota Formation as fossil evidence.

[b]Effectiveness is assessed from the literature, based on the electronic supplementary material, text S8 and mostly the electronic supplementary material, table S6, 'Florivory and pollination of modern basal angiosperms'. min, minimum; inter, intermediate; max, maximum.

[c]FFG; HF, hole feeding; SF, surface feeding; MF, margin feeding; and P&S, piercing-and-sucking. ND indicates no damage, or nondamagers.

[d]Data for the relative contribution of pollen versus nectar rewards comes principally from [49] and [50].

[e]For definitions and clade membership of mouthpart classes, see [36].

[f]See the electronic supplementary material for additional documentation.

[g]Tenebrionoidea includes Meloidae, Mordellidae and Tenebrionidae, listed in the electronic supplementary material, table S3.

[h]The clade Eulepidoptera contains the vast majority of moths and butterflies that are not otherwise mentioned in this section.

adjacent petal tissue. Damage type DT01 consists of holes 1 mm or less in diameter and is associated with flower morphotype 4 (electronic supplementary material, figure S5H–J). DT02 consists of holes between 1 and 5 mm in diameter and occurs on *Dakotanthus* (figure 1j–k). Dakota hole feeding consists of 21 perforations of DT01 and DT02 that represent 10.1% of all DT occurrences among flower morphotypes. Of all hole-feeding occurrences, 38.1% was present on *Dakotanthus cordiformis* from the Rose Creek I and II localities and 57.1% on flower morphotype 4 from the Braun's Ranch locality. It was noted that 71.4% of hole feeding was present on the lower half, rather than the upper half of the petals.

### (ii) Margin feeding
Margin feeding on Dakota Formation flowers is represented by DTs DT12, DT13 and DT405. DT12 and DT13 consist of cuspate to U-shaped excisions, typically several mm in chord length, occurring along the edges of petals and sepals (electronic supplementary material, text S4). The cut edge, in addition to a bordering rim of dark reaction tissue, occasionally displays micromorphological features such as protruding veinal stringers, necrotic tissue flaps and cuspules within the overall cut edge, analogous to damage on foliage. DT12 occurred along the petal side edges on *Dakotanthus* (figure 1c,h,i), flower morphotype 4 (electronic supplementary material, figure S5E,F), flower morphotype 5 (electronic supplementary material, figure S4A) and flower morphotype 17 (electronic supplementary material, figure S6F). DT13 was present on the tips of petals of *Dakotanthus* (figure 1a,b) and flower morphotype 5 (electronic supplementary material, figure S4A). Careful examination of DT13 was required to determine if the damage was present, to eliminate confusion with a retuse or apically embayed margin. DT405 is a newly described DT (electronic supplementary material, text S5) and previously has not been recorded in the fossil record.

### (iii) Surface feeding
The only example of a surface feeding FFG on a Dakota flower morphotype was DT DT29 (not illustrated) occurring on flower morphotype 7. DT29 is highly variable in size and shape, featuring polylobate to ovate patches of surface-fed petal tissue with distinct development of a reaction rim resulting from abrasion, scraping or delamination of a surface tissue layer (electronic supplementary material, text S4).

### (iv) Piercing-and-sucking
Piercing-and-sucking damage of Dakota floral morphotypes is represented by the five DTs of DT46, DT48, DT138, DT383 and DT402. These DTs consist of various patterns of punctures that penetrate or slice into shallow to deep floral tissues (electronic supplementary material, text S4). DT46 and DT48 are single, randomly dispersed punctures less than 1 mm in diameter, present on petals or other flower elements. DT46 is a circular, concave mark with a crater-like rim and occurs on *Dakotanthus* (figure 1c,e,j,l), flower morphotype 1 (electronic supplementary material, figure S5A,B), and flower morphotype 5 (electronic supplementary material, figure S5C,D,M–O). By contrast, rare DT48 (not illustrated) is an elliptical puncture, with either a cratered rim or a convex central boss. DT138 are linear rows of punctures that occur on flower morphotype 5 (electronic supplementary material, figure S4D,E). DT383 are compact circular to polylobate clusters of punctures that

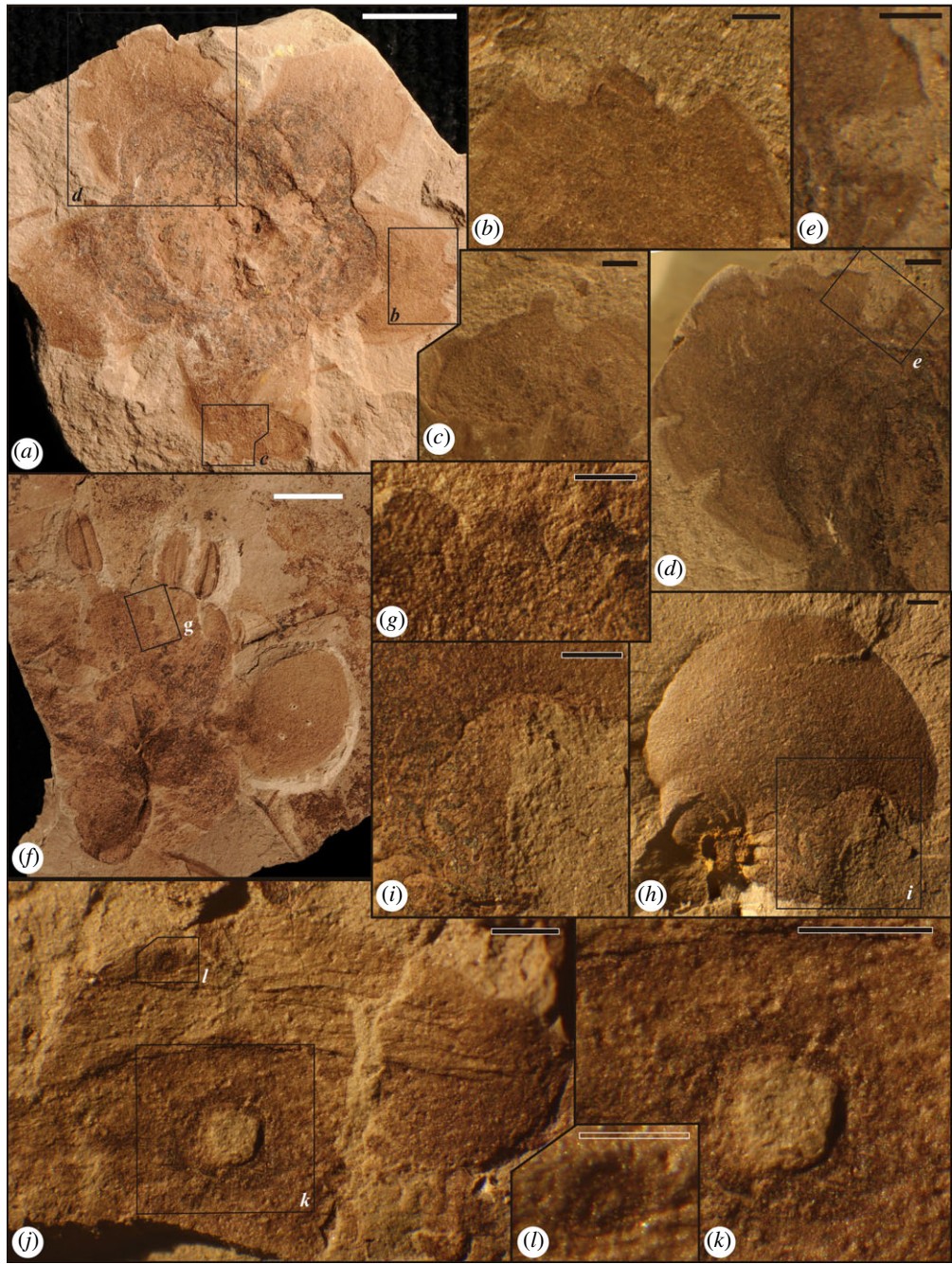

**Figure 1.** The florivore assemblage on *Dakotanthus cordiformis* (*a–l*), displaying petal DTs from all four FFGs of hole feeding, margin feeding, surface feeding and piercing-and-sucking. Specimen UF-12941 at (*a*) shows DT13 margin feeding cusps on the distal edge of the petal, enlarged in (*b*); smaller versions of DT12 margin feeding and DT46 punctures enlarged in (*c*). From the same specimen are several, V-shaped, margin feeding, DT405 notches along the petal edge, enlarged in (*d*), and further enlarged, including a DT46 ovate puncture, at (*e*). Specimen UF-3522 with several stamens at (*f*) shows a series of DT405 edge notches, enlarged in (*g*). Specimen UF-5612 displays DT12 margin feeding at the bottom of the petal at (*h*), enlarged at (*i*). *Dakotanthus cordiformis* (UF-5773) at (*j*) are DT02 hole-feeding damage, enlarged in (*k*), and, together with small dark, ovate punctures of DT46, enlarged in (*l*). Note well-developed reaction rim surrounding DT01. Scale bars: white, 5 mm; black, 1 mm; empty, 0.5 mm. (Online version in colour.)

probably accessed deeper tissues (electronic supplementary material, table S5). DT402, a newly described DT (electronic supplementary material, text S5), represents typically elongate, compact clusters of punctures in shallow tissues that occur especially along petal or sepal edges. DT402 occurs on flower morphotype 4 (electronic supplementary material, figure S5*F,G* and S5*K–M,O,P*) and flower morphotype 5 (electronic supplementary material, figure S4*B,C,F–I*). No preferential occurrences of piercing-and-sucking DTs were noted for the five DTs occurring on the upper versus lower halves of the petals.

## (f) Matching insect mouthpart classes with feeding damage

Associations were established between the pattern of Dakota insect damage with the relevant insect mouthpart class borne by an insect that would have produced that damage [36,52,53]. Such relationships, based on modern data [36] (electronic supplementary material, table S3), were made to better constrain the identities of potential florivores and pollinators (electronic supplementary material, text S6). However, Dakota damage caused by adult ectognathate,

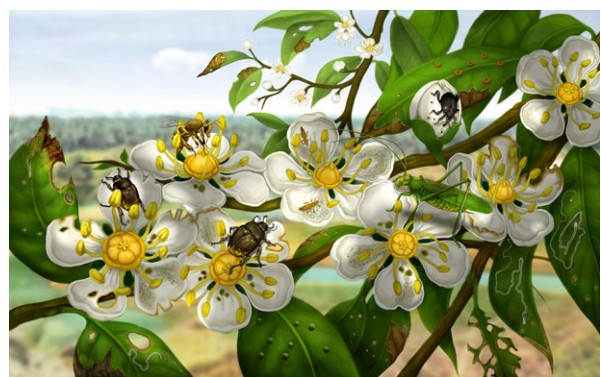

**Figure 2.** A reconstruction of the insect pollinator community on *Dakotanthus cordiformis* [7] based on patterns of florivory. This scene is from the Rose Creek locality of the Early Cretaceous (late Albian) Dakota Formation of Southwestern Nebraska, USA. Painted by Xiaoran Zuo. (Online version in colour.)

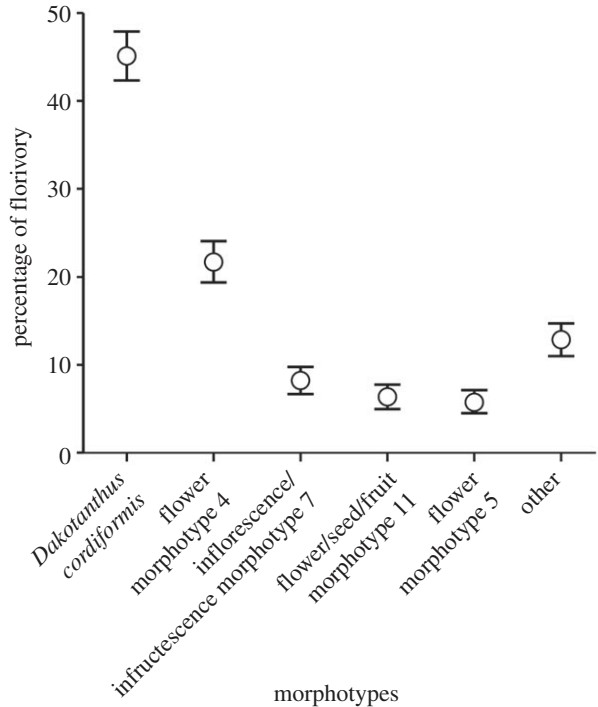

**Figure 3.** Percentage representation of folivory for the five most insect-damaged flower morphotypes.

larval ectognathate and sericterate mouthpart classes typically could not be separated from each other. These three mandibulate (chewing) mouthpart classes account for 37.7% of all DT occurrences. Damage caused by the maxillolabiate (a complex apparatus for nectar-extraction) and rhynchophorate (perforating) mouthpart classes that result in hole-feeding damage cannot be distinguished from each other, but these two mouthpart classes that create holes on leaves account for 10.1% of all DT occurrences. Damage caused by the segmented beak mouthpart class, responsible for puncturing deeper tissues with stylate mouthparts, involve piercing-and-sucking feeding and account for 31.4% of all DT occurrences. Damage attributable to mouthcone mouthparts modified for punch and sucking of shallow, epidermal tissues account for 20.8% of DT occurrences. These seven mouthpart classes, responsible for four major feeding styles, indicate that florivory was dominated by edge feeders and tissue penetrating piercer and suckers, and less so by hole feeders and shallowly penetrating piercer and suckers, reconstructed in figure 2.

## (g) Analyses of the damage

For the five most prevalent flower morphotypes, the two metrics expressing the abundance (DT occurrences) and percentage of DTs present are a near-exact match of each other (electronic supplementary material, table S4). This near duplication is shown in (i) the percentage contribution of each morphotype to the total, (ii) the morphotype rank order, and (iii) the cumulative totals. Much less similar is the third metric of the percentage of specimens that are florivorized, which departs from the two other metrics in exhibiting greater differences in the contribution of each morphotype to the total, a different morphotype rank order after the first two most florivorized morphotypes of *Dakotanthus* and flower morphotype 4, and a higher cumulative total of 97.7 versus the first two of 86.6 and 85.3.

A plot of the percentage of florivory of the three dominant FFGs of hole feeding, margin feeding and piercing-and-sucking was made for the major florivorized flower morphotypes of *Dakotanthus*, flower morphotype 4 and flower morphotype 5 for the three Dakota localities of Rose Creek I, Rose Creek II and Braun' Ranch (figures 3 and 4). This analysis revealed three patterns. First, the florivore–pollinator communities of flower morphotypes 4 and 5 at Braun's Ranch locality are

dominated by the piercing-and-sucking FFG, whereas the hole feeding and margin feeding FFGs played a minor role at both localities. By contrast, the *Dakotanthus* florivore faunas are dominated by the margin feeding FFG at the Rose Creek I and II localities, whereas the hole feeding and piercing-and-sucking FFGs have minor roles at both localities, although piercing-and-sucking appears to be subdominant at the Rose Creek II locality. Second, whereas margin feeding or piercing-and-sucking may have played a dominant role, depending on locality, hole feeding always had a minor role in the florivory spectrum across all localities. Third, with the exception of hole feeding, the florivore communities at Braun's Ranch versus the Rose Creek I and II localities, were largely feeding inversions of each other, suggesting heterogeneity in the pollinator assemblages by locality.

## 4. Discussion

### (a) How similar is Dakota florivory, folivores and pollinators to modern counterparts?

The presence of four FFGs subsuming 11 DTs in Dakota Formation plants (figures 1 and 2; electronic supplementary material, figures S4–S6) provides a very modern cast to the documented florivory (electronic supplementary material, table S3, text S4 and figure S7). In terms of the distinctiveness of the insect-mediated damage, the mouthpart classes responsible for the damage (electronic supplementary material, text S6), the proportional distribution of the damage on the flower morphotypes (electronic supplementary material, table S4), and their near-identical comparison to damage produced by known modern taxa [8] affirms the hypothesis that Dakota florivory is indistinguishable from its modern equivalent. This similarity suggests a similar pollinator community,

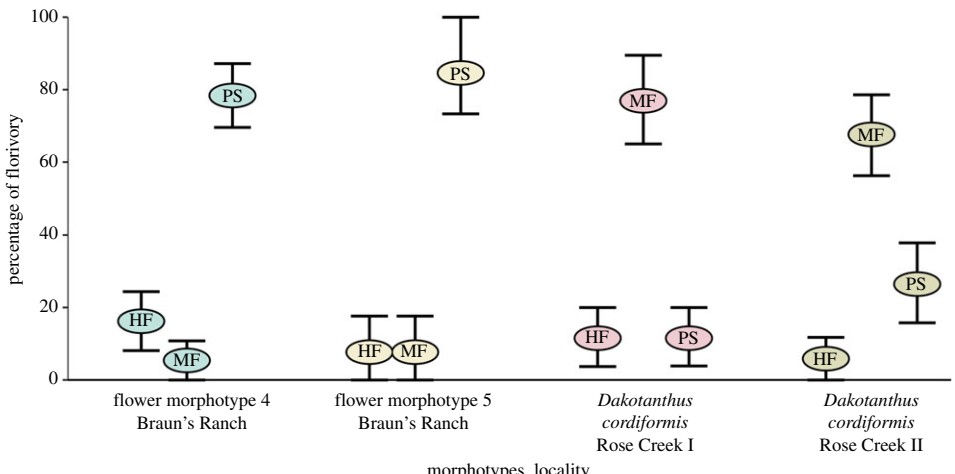

**Figure 4.** Per cent representation of florivore FFGs for flower morphotypes and sites. HF, hole feeding; MF, margin feeding; SF, surface feeding; PS, piercing-and-sucking. (Online version in colour.)

although evidence for fluid feeding, nondamaging adult taxa is circumstantial.

The richness of Dakota florivory and suggested associated pollinators provides, to our knowledge, the first extensive evidence for a community of insect visitors on the earliest, well-documented bowl-shaped flowers (electronic supplementary material, table S5, figure S7 and text S7). The florivore component of Dakota insect pollinators is established based on (i) distinctive DTs, (ii) extant lineages that were present during Dakota time from fossil occurrences, and (iii) evidence of relevant fossil occurrences or presence of closely related clades (phylogenetic bracketing) (electronic supplementary material, texts S6 and S7). The florivore assemblage consisted of a variety of insects with mandibulate and piercing-and-sucking mouthparts that produced recognizable damage patterns on flowers (electronic supplementary material, text S6). This pattern affirms the hypothesis that major taxonomic groups of insect florivores were very similar to their extant counterparts (electronic supplementary material, table S6 and text S8). Probably associated with this community of insect damagers of floral tissue, the indirect evidence indicates presence of lineages of fluid feeding, adult and nondamager taxa that left no trace on Dakota flowers. Based on the Dakota florivory data and modern studies, the core pollinators were heteropterans (especially pentatomorphs), thrips, polyphagan beetles (principally scarab and leaf beetles, and weevils), and bees. A subordinate component of early-diverging moths, sawflies, several major lineages of nematoceran and brachyceran flies, and perhaps parasitoid wasps probably were present (electronic supplementary material, table S6 and text S7).

## (b) Is there evidence for nectar robbing?

The data on hole feeding across the flower morphotype hosts is intriguing. Hole feeding consisting of 23 perforations of DTs DT01 and DT02 represent 10.1% of all DT occurrences among Dakota flower morphotypes. Of hole-feeding occurrences, 95.2% were present on *Dakotanthus* from the Rose Creek I locality and on flower morphotype 4 from the Braun's Ranch locality. There is a distinct preference for the lower half (71.4%) rather than the upper half of the petal. All holes were circular or nearly so, and no evidence of slits or tears was observed as holes on the petals. Although these data suggest that nectar robbing was present, nectar

robbing would be ineffectual if flowers are open, bowl shaped, and with rewards such as nectar readily available to insect visitors [54]. All modern nectar robbing occurs with tubular or similar flowers that that are highly enclosed and with access through a narrowly throated corolla [54]. The floral morphology of Dakota flowers, exemplified by *Dakotanthus*, is inconsistent with nectar robbing, and holes are circular and rounded in contrast with modern nectar robbers that construct slits or holes with jagged outlines [55,56]. Consequently, hypothesis 3, that nectar robbing was present on Dakota flowers, is rejected. Given this outcome, a more productive search for the earliest nectar robbing would be among early occurrences of tubular or otherwise enclosed flowers in the younger Late Cretaceous [2].

## (c) A brief historical perspective on Cretaceous angiosperm pollination

The early fossil history of angiosperm pollination [2] is illustrated by the bowl and similarly shaped floras in the 21 localities listed in geochronological order that provide data on inferred insect pollinator lineages and their functional-feeding group membership (electronic supplementary material, table S5 and text S7). This list shows that the Dakota insect pollinator fauna probably had taxonomic similarities to that of extant basal angiosperms (electronic supplementary material, table S6; texts S7 and S8). The next, four-million year-younger assemblage of pollinating insects originates from the very geographically, ecologically and taphonomically different locality of Myanmar amber, which shows a distinct pollinator fauna (electronic supplementary material, table S5 and figure S8). Subsequent, Late Cretaceous floras originate from a variety of localities that reveal the expansion of dicot angiosperms and an associated pollinator fauna.

## 5. Conclusion

This study provides a new approach for the study of pollination in the fossil record. By examining exquisitely preserved insect damage on Dakota flowers, the three hypotheses have been tested that were initially proposed in this study. The first hypothesis—*florivore damage patterns on Dakota flowers are similar to those of today*—is supported. One proviso to this

conclusion is that evidence for pollinating insects on Dakota flowers result from insect immatures and adults capable of leaving detectable damage, principally those with mandibulate and stylate mouthparts or their modifications, but excludes adult fluid feeding insects whose mouthparts are incapable of damaging tissues. Nevertheless, the indirect evidence of phylogenetic bracketing indicates the existence of these nondamaging, fluid feeding and pollinating clades. The second hypothesis—*major taxonomic groups Dakota insect florivores are similar to those of today*—is supported. Again, a caveat is that major insect taxa lacking the ability to inflict damage on flowers remain undetected. Circumstantial evidence based on fossil occurrences and phylogenetic bracketing indicates that typical nectar-feeding taxa were present. Last, the third hypothesis—*nectar robbing was present*—is rejected. The reason for rejection is not that a great preponderance of hole feeding was located at petal bases, but rather the open, bowl shape of Dakota flowers would obviate the need for nectar robbing. While addressing these three hypotheses extends understanding of florivory and pollination to the Dakota Formation, important gaps in knowledge of early angiosperm pollination remain. Highly relevant deposits from this time interval (electronic supplementary material, table 5, text S7 and figure S7) should be further explored.

Data accessibility. The data supporting the analyses of this article consist of (i) table 1 in this article; (ii) more extensive, linked electronic supplementary material that includes raw and related data among the three fossil localities as well as modern data; (iii) appendix 1 that provides detailed descriptions of flower morphotypes/species; and (iv) appendix 2 that provides R code files and Excel data files in a zip file. The data are provided in the electronic supplementary material.

Authors' contributions. L.X.: conceptualization, data curation, formal analysis, investigation, methodology, supervision, validation, visualization, writing—original draft, writing—review and editing; C.L.: conceptualization, funding acquisition, methodology, project administration, supervision, validation, visualization, writing—original draft, writing—review and editing; D.D.: conceptualization, data curation, investigation, resources, supervision, validation, visualization, writing—original draft, writing—review and editing; D.R.: conceptualization, data curation, funding acquisition, project administration, resources, supervision, validation, visualization, writing—original draft, writing—review and editing. All authors gave final approval for publication and agreed to be held accountable for the work performed therein.

Competing interests. We declare that we have no competing interests.
Funding. This research was supported by the National Natural Science Foundation of China (grant nos. 31730087, 32020103006 and 41688103). Fieldwork was supported in part by National Science Foundation of United States grant nos. DEB 75-02268, DEB 75-19849, DEB 77-04846, DEB 10720 and EAR 79-00898 to D.D.
Acknowledgements. We especially thank two reviewers who give us important comments. We thank Gussie Maccracken, Terry Lott, Finnegan Marsh, Steven Manchester, Hongshan Wang, Nareerat Boonchai and Xiaodan Lin who individually provided support in the Labandeira and Manchester laboratories. Sandra Schachat is thanked for producing figures 3 and 4. Gussie Maccracken provided a critique of a preliminary draft of the manuscript and offered needed overall advice. We are especially grateful to Steven Manchester for sponsoring the senior author at the Florida Museum of Natural History, in Gainesville for two months and their dedicated staff for general assistance. We thank Huayan Chen who provided photos of modern florivores and Torsten Dikow who identified the fly in electronic supplementary material, figure S6. We thank Liang Chen who helped us in figure construction. The Fossilworks database is acknowledged. This is contribution 378 to the Evolution of Terrestrial Ecosystems consortium at the National Museum of Natural History, in Washington, D.C.

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
