## [Peer Review File · Proceedings of the Royal Society B: Biological Sciences]

Review History

RSPB-2021-0320.R0 (Original submission)

Review form: Reviewer 1 (Angela Moles)

Recommendation

Major revision is needed (please make suggestions in comments)

Scientific importance: Is the manuscript an original and important contribution to its field?

Excellent

General interest: Is the paper of sufficient general interest?

Excellent

Quality of the paper: Is the overall quality of the paper suitable?

Acceptable

Is the length of the paper justified?

No

Should the paper be seen by a specialist statistical reviewer?

No

Do you have any concerns about statistical analyses in this paper? If so, please specify them explicitly in your report.

No

It is a condition of publication that authors make their supporting data, code and materials available - either as supplementary material or hosted in an external repository. Please rate, if applicable, the supporting data on the following criteria.

Is it accessible?

Yes

Is it clear?

Yes

Is it adequate?

Yes

Do you have any ethical concerns with this paper?

No

Comments to the Author

I think this is a very valuable piece of work. I have three main suggestions that I think will substantially improve the manuscript:

1) ALIGN YOUR TITLE/CONCLUSION AND YOUR EVIDENCE

Your title and conclusion suggest that you have studied pollination. What you really gathered data on was folivory. Your data provides lovely strong evidence for early association between flowers and invertebrates. I understand that there is a likely link between this and pollination, but that is not empirically established here. Thus, while I absolutely agree that you should make the link to many of these invertebrates likely being pollinators in the discussion, I think the rest of the paper (especially the title and conclusion, but also the introduction) need to be changed to accurately reflect the questions asked and the evidence presented. If you want to make conclusions about the relationship between pollinators and florivores, you need to do more to establish that link within the paper.

2) ALIGN YOUR RESULTS AND INTRODUCTION

Your main questions were: "First, what are the patterns of florivory and are these patterns similar to those made by modern florivores (text S1 of the electronic supplementary material)? Second, which insect visitors may have been responsible for florivory and closely related pollination that targeted Dakota flowers? Third, is there evidence of nectar robbing, or pollinator "cheating", among the fossil flowers?"

Your results section 1) describes the flower morphotype abundance and diversity, 2) describes damage patterns, 3) assesses host specificity, 4) describes the geographic distribution of florivory, 5) steps through the different damage types on different flower morphotypes, 6) matches insect mouthparts with feeding damage (which actually addresses the second question), and 7) quantifies the amount of folivory in each damage class.

That is, there is a striking mismatch between what you set out to do and what you actually did. I think you need to either answer the questions you introduce (in the main text, not the SI, where most readers will not see it), OR introduce the questions you answer so that the reader understands why the information you present is interesting and important and novel. I

personally would like to see the former approach (use the three questions as subheadings), but either approach would work.

3) REDUCE THE LENGTH OF THE RESULTS SECTION

The results section is remarkably long, and you consistently give more detail in the results text than is necessary. The current level of detail makes it harder for your readers to get the main points. I think you could greatly improve the paper by moving details that the average reader doesn't need (e.g. the detailed distribution of specimen types, and information that doesn't actually address the questions you set out to answer – such as detailed information on the different damage types on each flower morphotype) to the supplementary materials.

MINOR COMMENTS

1. Abstract, line 8 – do you actually have evidence for flies, moths and bees, or is this speculation? If the latter, it is appropriate to mention it in the discussion, but not to include it in the abstract.
2. Your use of acronyms makes the paper unnecessarily difficult to read. Can you please write out damage type and functional feeding group in full at each use?
3. Remove the (DT) information from the abstract – we do not need to know how many damage types you had in each category until the methods.
4. Page 6, 2nd line down, and elsewhere. This is another example of too much information. We don't know what DT46 and DT402 are – and we don't NEED to know this level of detail to understand the answers to your main questions. Put this level of information in the supplementary information, or possibly a table in the main text if you think it is really important – and keep the results text meaningful for your readers (ie use words not damage type codes).
5. Page 6, 10th line down. Significant is a bit of a loaded word in science writing. Do you mean not statistically significant, or negligible?
6. On the 20th line down on page 5, you state “Two noticeable damage patterns, discussed below, are (1), certain flower morphotypes that were most florivorized; and (2), particular FFGs and DTs that had the greatest abundance at localities.” This statement gives us almost no information. Of course some flower morphotypes got the most damage (that HAD to happen), and the second statement is completely without information.
7. I suspect you tried to hide the length of this paper by using small font, not double-spacing the text and avoiding line numbers. However, these choices all made it harder to review. Can we have it in standard manuscript format next time please?
8. The figures are very low-resolution. Make sure the figures align with the intro and results also.

Sorry to be so hard on you – I think this is a really really interesting topic and some fantastic data – I just think the paper could be a lot more effective in communicating the great things you found!

I do hope some of these comments are helpful,

Angela Moles

Review form: Reviewer 2 (David Wagner)

Recommendation

Accept with minor revision (please list in comments)

Scientific importance: Is the manuscript an original and important contribution to its field?

Excellent

General interest: Is the paper of sufficient general interest?

Good

Quality of the paper: Is the overall quality of the paper suitable?

Excellent

Is the length of the paper justified?

Yes

Should the paper be seen by a specialist statistical reviewer?

No

Do you have any concerns about statistical analyses in this paper? If so, please specify them explicitly in your report.

No

It is a condition of publication that authors make their supporting data, code and materials available - either as supplementary material or hosted in an external repository. Please rate, if applicable, the supporting data on the following criteria.

Is it accessible?

Yes

Is it clear?

Yes

Is it adequate?

Yes

Do you have any ethical concerns with this paper?

No

Comments to the Author

See attached. (See Appendix A)

Decision letter (RSPB-2021-0320.R0)

23-Mar-2021

Dear Ms Xiao:

Your manuscript has now been peer reviewed and the reviews have been assessed by an Associate Editor. The reviewers' comments (not including confidential comments to the Editor) and the very specific and constructive comments from the Associate Editor are included at the end of this email for your reference. As you will see, the reviewers and the Editors have raised

some concerns with your manuscript and we would like to invite you to revise your manuscript to address them.

Research ethics:

Use of animals and field studies:

It is a condition of publication that you make available the data and research materials supporting the results in the article. Please see our Data Sharing Policies (<https://royalsociety.org/journals/authors/author-guidelines/#data>). Datasets should be deposited in an appropriate publicly available repository and details of the associated accession number, link or DOI to the datasets must be included in the Data Accessibility section of the article (<https://royalsociety.org/journals/ethics-policies/data-sharing-mining/>). Reference(s) to datasets should also be included in the reference list of the article with DOIs (where available).

Please submit a copy of your revised paper within three weeks. If we do not hear from you within this time your manuscript will be rejected. If you are unable to meet this deadline please let us know as soon as possible, as we may be able to grant a short extension.

Best wishes,
Dr Maurine Neiman
<mailto:proceedingsb@royalsociety.org>

Associate Editor
Comments to Author:

We have received two extensively detailed and helpful reviews that provide many useful suggestions, which should be considered and addressed one point at a time.

There is a consensus amongst the three of us that the title is too broad and does not convey the main topic of the paper, which is damage to flowers. The title should probably mention the word "flowers" (instead of the jargon "florivory") and probably not "pollinated".

The first reviewer focuses on logic, organisation, and presentation. Her section 2 suggests that the questions raised in the introduction could be used to structure the rest of the paper. I think there are questions about the questions (see below). Section 3 suggests moving text to the supplement; I offer specific suggestions below. Questions about the abstract in particular are raised.

The second reviewer offers numerous specific suggestions and makes four broad ones. (1) The study should be put in the context of a time-dated phylogeny of the major holometabolous insect groups. Doing so would make it possible to test hypotheses: given that groups X, Y, and Z pollinate and existed by the latest Early Cretaceous, is there evidence that they left damage on flowers? If yes, what does that say about the rate of evolution of these interactions? If no, why not? (2) The paper talks about pollination a lot, but the actual evidence pertains to florivory, so the two need to be disentangled clearly and consistently. (3) The terms monophagy, oligophagy, and polyphagy are misapplied. (4) The evidence for nectar robbing per se is not very convincing, so that discussion could be cut or downplayed instead of being one of the three major hypotheses of the paper. The reviewer also offers numerous comments on the text that should be taken seriously.

My comments are as follows.

The abstract needs substantial work. Right now, it only discusses data, methods, and results up to the last sentence, which is vague and unlikely to grab reader attention. Many of the details should be omitted. The abstract should lead with a strong couple of sentences about the broader context of the study and the knowledge gap to be addressed. The last sentence or two should cycle back to the knowledge gap and explain how it has been filled. This gap may be along the lines of "did modern polinators and flower predators evolve into these roles right away after the first appearance of large flowers"?

The first paragraph of the introduction is fine, but the second paragraph focuses on descriptive questions that are not raised again in a clear, simple way in the conclusions. "What are" and "which" questions are not really hypothesis-testing questions, but instead purely descriptive. The authors should have a hard think about what hypotheses they are testing and how to explain them here. What are the alternatives, and how will they be distinguished?

The "conservative estimate" of species richness could be backed up by a quantitative statistic. There are many ways to do this. Plant ecologists often favour Fisher's alpha, which has advantages. I went ahead and computed alpha based on the counts in Fig. S2, which are:

292,140,53,41,36,16,13,5,5,4,4,4,3,3,2,2,2,2,1

I used the following script to compute alpha. It is based on an equation given by May (1975).

```
fisher<-function (n)
{
  S <- length(n)
  N <- sum(n)
  a <- 1
  lasta <- 0
  z <- 0
  while (abs(a - lasta) > 1e-07 && z < 1000) {
    z <- z + 1
    lasta <- a
    a <- S/log(1 + N/a)
  }
  x = N/(N + a)
  return(a)
}
```

I obtained 8.23, which is high but not very high based on tree inventory data in the Ecological Register:

http://ecoregister.org/eco.pl?a=getPlaces&life_forms=trees&displayed_rows=25&order_by=fisher&direction=descending

An alpha of 8 is roughly indicative of a tropical dry forest, and higher than anything you would see in the temperate zone.

There may be an error where the text mentions 645 specimens, because the above list adds up to 646.

The text says that Fig. S3 suggests a log normal distribution, but this is impossible to tell without logging the y-axis. Also, the log series is hard to distinguish visually from the log normal without a formal analysis. I went ahead and computed the fit of the counts to those two distributions using the fitLogNormal and fitLogSeries functions provided in the supplement to my 2015

Science Advances paper, and the data do in fact better fit the log normal. But the text should not say this without providing evidence.

The paragraph at the end of section 3a could be cut, as Fig. S2 gives the actual counts. The third and fourth paragraphs of 3b could be moved to the supplement or compacted.

Section 3e is very detailed and could be moved to the supplement.

Section 4a doesn't address a hypothesis and is descriptive. It possibly could be moved out or compacted.

Section 4b does address the third question in the introduction, but the second reviewer argues that the inference is unsteady, so this section needs work.

Section 4c also doesn't address a question, as currently written. It is somewhat confusing because bowl-shaped flowers dominate the authors' data set, but other kinds of flowers show up later and demonstrate pollinator associations. It's not clear why they wouldn't. So this section needs to either be cut, compacted, or made more interesting by referencing a yes/no hypothesis.

The conclusion seems to dance around a key point, which is that the paper is exciting exactly because no similar study exists at all and because the study was made possible by the really unusual preservation of this flora. This kind of information should have been highlighted in the abstract and also towards the end of the introduction, where readers would expect to see it. So the section could possibly be moved up. A new concluding paragraph focusing on what hypothesis was tested and how a knowledge gap was filled would be useful to see.

Board Member: 2

Comments to Author(s):

The manuscript presents a data set of apparently high palaeobiological importance, and it is apparent that there are no comparable studies, the samples are highly unusual, and much has been done to document them. However, much more could be done to put the work in context, especially in the abstract, introduction, and conclusions. The stated hypotheses are also narrowly focused, and should be recast to emphasise the contrasting possibilities that extensive folivory interactions did or did not evolve at such an early stage of angiosperm evolution, which are referenced in the title.

Reviewer(s)' Comments to Author:

Referee: 1

Comments to the Author(s)

I think this is a very valuable piece of work. I have three main suggestions that I think will substantially improve the manuscript:

1) ALIGN YOUR TITLE/CONCLUSION AND YOUR EVIDENCE

Your title and conclusion suggest that you have studied pollination. What you really gathered data on was folivory. Your data provides lovely strong evidence for early association between flowers and invertebrates. I understand that there is a likely link between this and pollination, but that is not empirically established here. Thus, while I absolutely agree that you should make the link to many of these invertebrates likely being pollinators in the discussion, I think the rest of the paper (especially the title and conclusion, but also the introduction) need to be changed to accurately reflect the questions asked and the evidence presented. If you want to make conclusions about the relationship between pollinators and florivores, you need to do more to establish that link within the paper.

2) ALIGN YOUR RESULTS AND INTRODUCTION

Your main questions were: “First, what are the patterns of florivory and are these patterns similar to those made by modern florivores (text S1 of the electronic supplementary material)? Second, which insect visitors may have been responsible for florivory and closely related pollination that targeted Dakota flowers? Third, is there evidence of nectar robbing, or pollinator “cheating”, among the fossil flowers?”

Your results section 1) describes the flower morphotype abundance and diversity, 2) describes damage patterns, 3) assesses host specificity, 4) describes the geographic distribution of florivory, 5) steps through the different damage types on different flower morphotypes, 6) matches insect mouthparts with feeding damage (which actually addresses the second question), and 7) quantifies the amount of folivory in each damage class.

That is, there is a striking mismatch between what you set out to do and what you actually did. I think you need to either answer the questions you introduce (in the main text, not the SI, where most readers will not see it), OR introduce the questions you answer so that the reader understands why the information you present is interesting and important and novel. I personally would like to see the former approach (use the three questions as subheadings), but either approach would work.

3) REDUCE THE LENGTH OF THE RESULTS SECTION

The results section is remarkably long, and you consistently give more detail in the results text than is necessary. The current level of detail makes it harder for your readers to get the main points. I think you could greatly improve the paper by moving details that the average reader doesn't need (e.g. the detailed distribution of specimen types, and information that doesn't actually address the questions you set out to answer – such as detailed information on the different damage types on each flower morphotype) to the supplementary materials.

MINOR COMMENTS

1. Abstract, line 8 – do you actually have evidence for flies, moths and bees, or is this speculation? If the latter, it is appropriate to mention it in the discussion, but not to include it in the abstract.
2. Your use of acronyms makes the paper unnecessarily difficult to read. Can you please write out damage type and functional feeding group in full at each use?
3. Remove the (DT) information from the abstract – we do not need to know how many damage types you had in each category until the methods.
4. Page 6, 2nd line down, and elsewhere. This is another example of too much information. We don't know what DT46 and DT402 are – and we don't NEED to know this level of detail to understand the answers to your main questions. Put this level of information in the supplementary information, or possibly a table in the main text if you think it is really important – and keep the results text meaningful for your readers (ie use words not damage type codes).
5. Page 6, 10th line down. Significant is a bit of a loaded word in science writing. Do you mean not statistically significant, or negligible?
6. On the 20th line down on page 5, you state “Two noticeable damage patterns, discussed below, are (1), certain flower morphotypes that were most florivorized; and (2), particular FFGs and DTs that had the greatest abundance at localities.” This statement gives us almost no information. Of course some flower morphotypes got the most damage (that HAD to happen), and the second statement is completely without information.

7. I suspect you tried to hide the length of this paper by using small font, not double-spacing the text and avoiding line numbers. However, these choices all made it harder to review. Can we have it in standard manuscript format next time please?

8. The figures are very low-resolution. Make sure the figures align with the intro and results also.

Sorry to be so hard on you – I think this is a really really interesting topic and some fantastic data – I just think the paper could be a lot more effective in communicating the great things you found!

I do hope some of these comments are helpful,

Angela Moles

Referee: 2

Comments to the Author(s)

See attached

Author's Response to Decision Letter for (RSPB-2021-0320.R0)

See Appendix B.

Decision letter (RSPB-2021-0320.R1)

28-Apr-2021

Dear Ms Xiao

I am pleased to inform you that your manuscript RSPB-2021-0320.R1 entitled "Florivory of Early Cretaceous flowers by functionally diverse insects: Implications for early angiosperm pollination" has been accepted for publication in Proceedings B.

The referee(s) have recommended publication, but also suggest some minor revisions to your manuscript. Therefore, I invite you to respond to the referee(s)' comments and revise your manuscript. Because the schedule for publication is very tight, it is a condition of publication that you submit the revised version of your manuscript within 7 days. If you do not think you will be able to meet this date please let us know.

When submitting your revised manuscript, you will be able to respond to the comments made by the referee(s) and upload a file "Response to Referees". You can use this to document any changes you make to the original manuscript. We require a copy of the manuscript with revisions made

since the previous version marked as 'tracked changes' to be included in the 'response to referees' document.

Sincerely,
Dr Maurine Neiman
Editor, Proceedings B
<mailto:proceedingsb@royalsociety.org>

Associate Editor:

Board Member

Comments to Author:

RSPB-2021-0320.R1 constitutes a major revision of the text that is thorough and sincere. The revision and extraordinary cover letter address all major and minor issues that were raised. However, a few minor problems remain.

There is some confusion as to who wrote the cover letter, which is signed by Dong and Labandeira but starts out by saying it is being submitted by Xiao. Regardless, the tracked changes indicate that Labandeira has done the heavy lifting. I think the author contributions statement is still accurate, though.

The new title is okay, but "Florivory" is jargon and echoes "flowers" too strongly. "Damage" or maybe "Consumption" would be fair alternatives.

The abstract is still descriptive in tone, mentioning a knowledge gap in a general way but not suggesting a hypothesis. It also doesn't explain the unusual nature of the floral assemblage compared to everything else in the palaeobotanical record, which is arguably the main point of excitement about the paper. I earlier suggested mentioning a question such as whether "modern pollinators and flower predators [evolved] into these roles right away after the first appearance of large flowers". This sort of thing could still be worked in.

The new second half of the introduction is fine, but belabours the "hypothesis" language. Some trimming of the "alternative hypothesis" sentences is warranted.

Reviewer 2 wanted to see a time-dated phylogeny of the major holometabolous insect groups. No specific step in this direction has been taken, but the supplement now includes lengthy discussions of the roles and first appearances of insect groups, and I think this is sufficient.

The "conservative estimate" of species richness based on uncertain reasoning remains unchanged, despite the fact that the authors could present their own, new estimate, as suggested. Specifically, it would be a good idea to briefly mention the Fisher's alpha figure for this data set in the main body of text (it is 8.226521, as can be confirmed by using the R library `sads`). The reason is that plant ecologists will immediately grasp the meaning of this number, instead of wondering where the broad range comes from and what it might mean in a standard ecological context.

There appears to have been a mixup with respect to Fig. S3. The y-axis of this figure needs to be logged, and the response letter says it has been. However, the original figure remains in the supplemental file. This is a serious problem because the text continues to suggest that the distribution is log normal, which can't be supported based on such a representation of the data. I do believe that the distribution does not follow the log series and is instead close to log normal, but readers should be able to see the evidence in graph form.

Despite the above, I congratulate the authors on a job well done.

Author's Response to Decision Letter for (RSPB-2021-0320.R1)

See Appendix C.

Decision letter (RSPB-2021-0320.R2)

06-May-2021

Dear Ms Xiao

I am pleased to inform you that your manuscript RSPB-2021-0320.R2 entitled "Florivory of Early Cretaceous flowers by functionally diverse insects: Implications for early angiosperm pollination" has been accepted for publication in Proceedings B.

The referee(s) have recommended publication, but also suggest some minor revisions to your manuscript. Therefore, I invite you to respond to the referee(s)' comments and revise your manuscript. Because the schedule for publication is very tight, it is a condition of publication that you submit the revised version of your manuscript within 7 days. If you do not think you will be able to meet this date please let us know.

Online supplementary material will also carry the title and description provided during submission, so please ensure these are accurate and informative. Note that the Royal Society will

not edit or typeset supplementary material and it will be hosted as provided. Please ensure that the supplementary material includes the paper details (authors, title, journal name, article DOI). Your article DOI will be 10.1098/rspb.[paper ID in form xxxx.xxxx e.g. 10.1098/rspb.2016.0049].

Sincerely,

Dr Maurine Neiman

Associate Editor:

Comments to Author:

RSPB-2021-0320.R2 is in excellent shape and is basically ready to go. The only remaining issues are very minor: some typos; a sentence that needs to be split and rewritten; some missing information in one passage of the supplement; and the probable need to use a different R package to get a richness estimate, and also to get the fitted line that is shown in a supplemental figure. These are all trivial.

- There is a typo in the second sentence of the second paragraph of p. 3 ("inset" instead of "insect").
- "bulk flora involving foliage." (p. 5) should read "bulk flora involving foliage)."

- "D. Dilcher, S. Manchester [10,36], but mostly by L. Xiao" (p. 7) should read "D. Dilcher, S. Manchester [10,36], and L. Xiao, but mostly by the latter"
- "piercing and sucking." (p. 8) should read "piercing and sucking"
- "(Fisher's alpha =7.94) of the number of available hosts at each locality" (p. 9) should just read "of the number of available hosts at each locality" based on the following.
- "Dakotanthus" (near the top of p. 10) isn't capitalised.
- There's a stray comma in "hole-feeding damage, cannot be" (bottom of p. 11).
- Another in "[8], affirms the hypothesis" (p. 13).
- Another in "lower half (71.4%), rather than" (p. 14).
- Another in "on flowers, remain undetected" (p. 15).
- "indicates that the existence" should read "indicates the existence" (p 15).
- "adpression material such as" should read "adpression material, such as" (p. 16).
- The supplement refers to something called "fisher test" in R. The function `fisher.test` is unrelated to Fisher's alpha and has to do with likelihood ratios in contingency tables, not species diversity. I'm not sure what function was used. This language should be revised to make it clear that `fisher.test` was not used.
- The equations involving y_0 , x_c , w , and A coefficients in Fig. S3 appear to represent the log normal distribution (compare with the PDF for the log normal: https://en.wikipedia.org/wiki/Log-normal_distribution). This is okay because the distribution does seem to be log normal. However, it's not clear exactly how these coefficients were fit, because the equation isn't of a garden-variety linear model. Therefore, functions like GLM couldn't have been used. Exact R functions need to be named. I note that fitting the log normal is extremely tricky and is done badly by package `sads`; a much better fitting routine (`poilogMLE`) is provided by package `poilog`, which is very easy to use.
- The main body of text says "our estimate from a Fisher's alpha test is 645 species" but this figure actually should have come from fitting the log normal. Fisher's alpha has nothing to do with the log normal (instead, it's the governing parameter of the log series), and it does not provide a richness estimate per se. I ran the $N = 646$ count data set (I can't figure out which individual is missing in the final data set) through `poilogMLE`, and I got a completely different figure. There are 36 species in that list and the "p" completeness value given by `poilogMLE` is 0.1836883, so the richness estimate should be $36/0.1836883 = 195.98$. Therefore, the "conservative" estimate of 150-200 species is not too bad! Obviously, the estimate will come out a little different once that one-specimen correction is made.
- I do still think that Fisher's alpha should be mentioned because plant ecologists will get something out of it.
- Putting that together, I think the main body of text could say "Based on the diversity and abundance of floral morphotypes (figure S2), our estimate from fitting the log normal is 196 species. We also obtained a Fisher's alpha value of 7.94".

Thanks to the authors for their diligence.

Decision letter (RSPB-2021-0320.R3)

18-May-2021

Dear Ms Xiao

I am pleased to inform you that your Review manuscript RSPB-2021-0320.R3 entitled "Florivory of Early Cretaceous flowers by functionally diverse insects: Implications for early angiosperm pollination" has been accepted for publication in Proceedings B.

The referee(s) do not recommend any further changes. Therefore, please proof-read your manuscript carefully and upload your final files for publication. Because the schedule for publication is very tight, it is a condition of publication that you submit the revised version of your manuscript within 7 days. If you do not think you will be able to meet this date please let me know immediately.

To upload your manuscript, log into <http://mc.manuscriptcentral.com/prsb> and enter your Author Centre, where you will find your manuscript title listed under "Manuscripts with Decisions." Under "Actions," click on "Create a Revision." Your manuscript number has been appended to denote a revision.

You will be unable to make your revisions on the originally submitted version of the manuscript. Instead, upload a new version through your Author Centre.

1) A text file of the manuscript (doc, txt, rtf or tex), including the references, tables (including captions) and figure captions. Please remove any tracked changes from the text before submission. PDF files are not an accepted format for the "Main Document".

2) A separate electronic file of each figure (tiff, EPS or print-quality PDF preferred). The format should be produced directly from original creation package, or original software format. Please note that PowerPoint files are not accepted.

3) Electronic supplementary material: this should be contained in a separate file from the main text and the file name should contain the author's name and journal name, e.g. `authorname_procb_ESM_figures.pdf`

All supplementary materials accompanying an accepted article will be treated as in their final form. They will be published alongside the paper on the journal website and posted on the online figshare repository. Files on figshare will be made available approximately one week before the accompanying article so that the supplementary material can be attributed a unique DOI. Please see: <https://royalsociety.org/journals/authors/author-guidelines/>

4) Data-Sharing and data citation

It is a condition of publication that data supporting your paper are made available. Data should be made available either in the electronic supplementary material or through an appropriate repository. Details of how to access data should be included in your paper. Please see <https://royalsociety.org/journals/ethics-policies/data-sharing-mining/> for more details.

If you wish to submit your data to Dryad (<http://datadryad.org/>) and have not already done so you can submit your data via this link <http://datadryad.org/submit?journalID=RSPB&manu=RSPB-2021-0320.R3> which will take you to your unique entry in the Dryad repository.

Once again, thank you for submitting your manuscript to Proceedings B and I look forward to receiving your final version. If you have any questions at all, please do not hesitate to get in touch.

Sincerely,
Dr Maurine Neiman
Editor, Proceedings B
<mailto:proceedingsb@royalsociety.org>

Associate Editor

Comments to Author:

Again, there are no serious issues at this point. There is a stray word ("fit") in the sentence about "fitting the log normal" that I suggested adding. There is a typo in the supplement ("Fischer" should read "Fisher"). The function used to compute Fisher's alpha is modified from the one I provided, but there is no citation. The citation is Alroy, J. (2015). The shape of terrestrial abundance distributions. *Science Advances* 1:e1500082.

Decision letter (RSPB-2021-0320.R4)

20-May-2021

Dear Ms Xiao

I am pleased to inform you that your manuscript entitled "Florivory of Early Cretaceous flowers by functionally diverse insects: Implications for early angiosperm pollination" has been accepted for publication in Proceedings B.

Data Accessibility section

Open Access

Paper charges

Sincerely,

Proceedings B

Appendix A

The is a terrific summary of one of the most well-known fossil plant assemblages on the planet, one which the authors have been studying for decades. The details in the narrative reflect their deep knowledge of both the plants, paleoinsect fauna, and fossils. Between the text and the extensive supplementary documents, there is a wealth of data shared in this submission, which is reason enough to accept the manuscript. I must also admit that the collective breath (and depth) of the four authors is his humbling, and, more to the point, greatly exceeds what I could assess. Their approach is novel, i.e., using damage to flowers (by florivores) to demonstrate that there was a diverse and an abundant pollinator fauna in place the early Cretaceous association that they studied. In their words, “the study provides a new approach for the study of pollination in the fossil record.”

The narrative is excellent, focused, appropriately detailed, and clean. The submission is wonderfully illustrated. I do have several suggestions that would improve the clarity of presentation and put some of the findings in a more understandable or broader context. I also share a couple instances where my interpretation would differ from that of the authors, and an instance where I disagree with their use of terms (i.e., their definitions of monophagous, oligophagous, and polyphagous). Without knowing the species of the fossils, these terms are misapplied.

My single largest suggestion is that they make an effort to track down a time-dated phylogeny for Coleoptera, Diptera, Hymenoptera, and Lepidoptera to find out what flower-visiting insect families were present 100 mybp and do a better job of conveying what pollinators were present *that do not damage flowers*. And then include one solid paragraph that discusses such at the end of the Discussion to support their own title and thesis that “Early Cretaceous Angiosperms were Pollinated by a Functionally Diverse Insect Fauna.” The ones mentioned in the paper/discussion presently are a bit random and woefully incomplete. I am a lepidopterist so see obvious omissions from the discussion: micropterigids, adelids, choreutids, scythridids, sesiids, zygaenids, and many others. By 100 mybp most Lepidoptera superfamilies were present, so singling out Gelechiidae misses a bigger point. Maybe just suggest superfamilies with known pollinators? This effort/any listing need not be exhaustive. [Minor point: I would not equate Gelechiidae with leaf rollers (which are mostly tortricids and many gracillariids) as appears to be the case in the text from my reading. Gelechiidae are shelter formers...and few are folivores.] But this paragraph, if included, could mention sawflies, several lineages of nemtacerans (e.g., bibionids, culicids) and other flies, and likely a dozen or more families of beetles.

Related to above, I am not especially happy giving equal weight to bees and katydid in a single listing of potential pollinators, e.g., in last sentence of Discussion. I think the list of pollinators should be divided into two classes. Likely pollinators: bees, beetles, flies (not listed in Discussion sentence), moths, a few hemipteran families, and thrips; and then a list of the “also rans”: e.g., grasshoppers, crickets, katydids, whatever.

I would like to see the authors do more to disambiguate florivory from pollination. Their frequent juxtaposition in the manuscript suggests/implies a strong association when there need be none. Best to add a few sentences that convey that not all folivores are pollinators and not all pollinators are folivores, and that an important fraction of the floral damage treated in this manuscript may have little to do with pollination/pollinators. Caterpillars and katydids that do

not move between plants and consume whole flowers are bad actors in this system. Across the Lepidoptera there are many lineages with caterpillars that feed on flowers that do not pollinate: caterpillars of Lycaenidae, Stiriinae (Noctuidae), *Eupithecia*, Heliiothinae (Noctuidae), and flower-feeding cutworms (Noctuidae). (And were it me, I would downplay Orthoptera—they eat flowers from the outside in, often leave after consuming petals, and only rarely would be moving pollen between flowers/plants.

Somewhere in the text it might be good to admit that the timing of florivory is important. If the florivory happens before a flower opens and before the pollen/sexual tissue is mature, the florivory would equate to predation.

Strictly speaking, damage specificity can't reveal host specificity as defined (and used) in a voluminous ecological literature. The matter of diet breadth being inferred from damage is a stretch: a leap of faith if we don't know the relatedness of the hosts, which is unknown for this fossil assemblage. And using just three instances as your criterion is an exceeding arbitrary number if there are dozens of fossils for a given morphotaxon. What if the insect is a generalist but the ecosystem had only a single host in flower in bloom at the instant of the preservation event? Then, the florivore would appear to be a specialist when in fact it is not. Insects in the same genus or tribe might all make the same type of damage but have different hosts and be strictly monophagous, but these would appear to generalists. This is too loose for me to endorse and requires terms other than monophagy, oligophagy, and polyphagy. Perhaps add "apparent" to these words or find some other work around. I am not on board otherwise.

It is a reach and not very defensible to equate hole feeding with nectar robbing. Far more parsimonious (and ecologically common) interpretation for "hole boring" would be to gain entry into a flower when the flower was almost open but not yet. Most importantly, nectar robbing occurs with complex flower morphologies, where bees are excluded by the flower design, e.g., when nectar is hidden deep in the corolla. We have *open bowl-shaped flowers* here. Heck, just fly in and collect the pollen/nectar—why waste time boring through the petals? [FTR: Bees that are nectar thieves. generally make rough, jagged-edged holes; circular holes would like be assoc. with a beetle or a small caterpillar.]

Which reminds: Please add a paragraph somewhere about evidence for nectar being present in the various floral types. When do we/you first have clear evidence of nectar rewards in angiosperms? It is somewhat assumed/implied that nectar is available in these flowers. But some early flowers may have lacked nectar and primarily supplied pollen as the principal reward. Please discuss this matter. It would be of obvious importance to the taxonomic composition of guild of pollinators that would have been present. Many of the suggested/implicated pollinators here require a nectar reward, others on pollen, and some both. Might even be worth mentioning such in the Introduction.

Minor gripe: text conflates DTs damage types with damage. It appears that DT is used for two different meanings: (1) for the different types (there are ?11 recognized) and (2) for number of damaged specimens. Find a way to disentangle and clarify.

Please consider reversing primacy of your numbers for the ?11 different damage types and the

arbitrary numbers you have selected to designate these. The reader shouldn't have to flip to a table to figure out what kind of damage you are discussing. At least initially focus on a word description of the damage type and include your number in parentheses. At the very least it would make the text more interesting.

I have suggested a few edits in a Word version that I created from the pdf. The authors can draw from this what they want.

Overall, this is an excellent manuscript, and I was happy to get a chance to have an early look.

David Wagner

Appendix B

Capital Normal University
105 Xisanhuanbeilu, Hadian District
Beijing, 100048
P.R. China
15 April 2021

Dr. Maurine Neiman
Subject Editor,
Proceedings of the Royal Society B (Biological Sciences)

Dear Editor Neiman,

On behalf of coauthors Conrad Labandeira, David Dilcher, and Ren Dong, I (Lifang Xiao) am pleased to return our substantially revised manuscript with a new title, “Florivory of Early Cretaceous flowers by functionally diverse insects: Implications for early angiosperm pollination” (RSPB-2021-0320), to the *Proceedings of the Royal Society B (Biological Sciences)*. All of the issues have been addressed by the four reviewers of our previous version of the manuscript – Associate Editor, *Proceedings* Board Member 2, Reviewer 1 and Reviewer 2 –, enumerated below. We agree with almost all of the concerns raised by the Reviewers. We have also addressed issues regarding research ethics, conflicts of interest, use of animals in field studies, and data accessibility and citation. We believe the manuscript is in good shape for further consideration for publication in the *Proceedings*.

Should the manuscript be accepted, we are submitting a reconstruction of our conception of florivory on the most heavily used of the Dakota Formation flowers, *Dakotanthus cordiformis*. We suggest that this reconstruction could be used for the cover of the issue in which our contribution would appear, or perhaps a graphic abstract. Other uses of the reconstruction could be contemplated.

Submission Requirements

1. Data accessibility and data citation. It is a condition of publication that you make available the data and research materials supporting the results in the article. Please see our Data Sharing Policies (<https://royalsociety.org/journals/authors/author-guidelines/#data>). Datasets should be deposited in an appropriate publicly available repository and details of the associated accession number, link or DOI to the datasets must be included in the Data Accessibility section of the article (<https://royalsociety.org/journals/ethics-policies/data-sharing-mining/>). Reference(s) to datasets should also be included in the reference list of the article with DOIs (where available).

Authors’ response. We have made all data not present in the main text available through the Electronic Supplementary Materials.

2. Electronic supplementary material. All supplementary materials accompanying an accepted article will be treated as in their final form. They will be published alongside the paper on the journal website and posted on the online figshare repository. Files on figshare will be made available approximately one week before the accompanying article so that the supplementary material can be attributed a unique DOI. Please try to submit all supplementary material as a single file.

Author’s response. We have submitted the Electronic Supplementary Material as a single file.

Reviewer Associate Editor

3. Editor's comments. "Comments to Author: We have received two extensively detailed and helpful reviews that provide many useful suggestions, which should be considered and addressed one point at a time."

Authors' response. We have carefully gone through and addressed all suggestions, corrections, and comments by the Associate Editor, Reviewer Board Member 2, Reviewer 1 (Angela Moles), and Reviewer 2 (David Wagner).

4. Editor's comments: Title. There is a consensus amongst the three of us that the title is too broad and does not convey the main topic of the paper, which is damage to flowers. The title should probably mention the word "flowers" (instead of the jargony "florivory") and probably not "pollinated".

Authors' response. We have crafted a new title that succinctly summarizes the major thesis of our study: *Feeding on Early Cretaceous Flowers by Functionally Diverse Insects: Implications for Early Angiosperm Pollination*. We agree that "florivory" is a bit jargony but would like to keep the word "pollination" somewhere in the title, as the implications of florivory for pollination is a major consequence of our study.

5. Editor's comments: Manuscript organization. The first reviewer focuses on logic, organisation, and presentation. Her section 2 suggests that the questions raised in the introduction could be used to structure the rest of the paper. I think there are questions about the questions (see below). Section 3 suggests moving text to the supplement; I offer specific suggestions below. Questions about the abstract in particular are raised.

Authors' response. We have restructured the abstract. In addition, we have acted on the issue of framing this study in terms of three yes/no hypotheses, rather than three questions, in the second paragraph of the Introduction. We return to these three hypotheses the Conclusion Section, where we state whether these hypotheses have been supported or rejected. (Also see our answer to Point 7 below.)

6. Editor's comments: Major suggestions of Reviewer 2. The second reviewer offers numerous specific suggestions and makes four broad ones. (1) The study should be put in the context of a time-dated phylogeny of the major holometabolous insect groups. Doing so would make it possible to test hypotheses: given that groups X, Y, and Z pollinate and existed by the latest Early Cretaceous, is there evidence that they left damage on flowers? If yes, what does that say about the rate of evolution of these interactions? If no, why not? (2) The paper talks about pollination a lot, but the actual evidence pertains to florivory, so the two need to be disentangled clearly and consistently. (3) The terms monophagy, oligophagy, and polyphagy are misapplied. (4) The evidence for nectar robbing per se is not very convincing, so that discussion could be cut or downplayed instead of being one of the three major hypotheses of the paper. The reviewer also offers numerous comments on the text that should be taken seriously.

Authors' response. We have addressed these four general issues. We agree with the suggestions of Reviewer 2. First, in terms of placing the insect groups – nonholometabolous and holometabolous – in dated phylogenies, we refer the reader to Text S6 and Text S7 of the Electronic Supplementary Material. The bottom line here is that, as mentioned by Reviewer 2 for Lepidoptera in point 33 below, with few possible exceptions such as Apoidea, virtually all of these insect groups were present by Dakota Formation time, during the latest Albian at about 103 million years ago. (See our response to point 32 below.) So, these geochronological occurrences and timings are now buttressed by cited references in Text S6 and Text S7 of the Electronic Supplementary Material.

Second, for the disentangling of florivory from pollination, we have added a new paragraph that accomplishes this goal, addressed in Point 36 below. In the Materials and Methods Section, we have added new Subsection 2d ("Distinguishing folivores and pollinators"), as follows:

“(d) Distinguishing florivores and pollinators

Insect visitors to flowers are of two fundamental groups, florivores and pollinators [30]. Not all florivores are pollinators and not all pollinators are florivores, and the relationships between these two ecological guilds are complex [8]. Florivores typically leave damage on flowers, overwhelmingly on petals [31], often resulting in negative interactions [32]. However, some florivore interactions are neutral or even positive [31,33], as petals occasionally contain nutritive or highly scented tissues designed for consumption by florivores as pollinators [34,35]. Florivory can be a form of predation if plant embryonic tissues are destroyed before floral the opening of the flower, or if there is consumption of immature pollen, features that do not appear present in bowl-shaped Dakota flowers, as the damage is overwhelmingly on inner petal surfaces. Consequently, florivores such as Orthoptera (katydids), Hemiptera (aphids, bugs), Thysanoptera (thrips), Coleoptera (beetles), and Hymenoptera (sawflies, wasps, bees) with mandibulate, styletate, or similarly modified mouthparts [36], provide good proxy data for the broad spectrum of pollinator interactions on flowers [37] (Table 1). However, a substantial component such as most adult Diptera [38] and Lepidoptera are nondamagers, as they do not leave damage on flowers.”

Third, we have eliminated the terms “monophagy”, “oligophagy”, and “polyphagy”, which places the burden of proof on the insect side of the relationship, for which we have no evidence. Instead, we use the terms “host specialized damage”, “intermediate specialized damage”, and “host generalized damage”, based on the distribution of the DTs on the flower morphotypes. This now places evidence of targeting flower morphotypes on the plant side of the relationship, for which we have substantial evidence, such as the categorization of flower morphotypes and the distribution of DTs on those morphotypes. The relevant part of the text is in Subsection 3c (“Assessing florivore host specificity by flower morphotype”) of the Results Section:

“The distribution of damage types (DTs) on plant hosts revealed three levels of host specificity [23]. Borrowing from studies of fossil herbivory as an example [24], host specificity is categorized as specialized damage if three or more occurrences of the same DT are present on the same host morphotype or on a very closely related host; damage is of intermediate specificity if the distribution of three or more occurrences of the same DT are present on more distantly related hosts; and generalized damage if three or more occurrences of the same DT are present on unrelated hosts [23]. For Dakota folivory data, because the phylogenetic relationships among flower morphotypes are unknown, terms expressing host specificity are referenced to the distribution of DTs on the flower species and morphotypes (table S2). The three examples of specialist damage are small hole-feeding DT01 on Flower Morphotype 4 that hosts 12 of 13 (92.3%) of all occurrences; circular holes between 1–5 mm in diameter on *Dakotanthus* that hosts all 8 (100%) of occurrences; and notched margin feeding of DT405 along the petal edges on *Dakotanthus*, which hosts 66 of 67 (98.5%) of all occurrences. The single example of damage of intermediate specificity is DT12, evidenced by too few DT distributions across three flower morphotypes. Seven examples of generalized damage are present. They are single, random piercing-and-sucking DT46 on *Dakotanthus* and flower morphotypes 6, 9 and 10; and clustered piercing-and-sucking of DT402 on *Dakotanthus* and flower morphotypes 1, 5 and 8; and DT13, DT29, DT48, DT138 and DT383 that defaults to generalized specificity, each having one or two occurrences on *Dakotanthus* and flower morphotypes 4, 5 and 7. This pattern of host specificity indicates three examples of specialized damage, one of intermediate specificity damage, and seven of generalized damage (Table S2).”

Fourth, we agree that the evidence for nectar robbing is tenuous at best, Consequently, we rejected our third hypothesis that “*Nectar robbing was present on Dakota flowers*”, mentioned in the second

paragraph of our Introduction Section. We have recast Subsection 4b of the Discussion Section, discussing the evidence regarding nectar robbing. In the Conclusion Section, we state that “[L]ast, the third hypothesis – *nectar robbing was present* – is rejected.” Reviewer 2 made the astute observation that nectar robbing pertains to tubular flowers where access to nectar is difficult for insects on the outside. By contrast, and nectar robbing is a useless endeavor when flowers are open, bowl shaped, and nectaries are readily accessible to florivores or pollinators. The fact that about 70% of holes (DT1 and DT2) occurred at the bases of petals initially threw us off.

7. Editor’s comments: Abstract. “My comments are as follows. The abstract needs substantial work. Right now, it only discusses data, methods, and results up to the last sentence, which is vague and unlikely to grab reader attention. Many of the details should be omitted. The abstract should lead with a strong couple of sentences about the broader context of the study and the knowledge gap to be addressed. The last sentence or two should cycle back to the knowledge gap and explain how it has been filled. This gap may be along the lines of “did modern polinators and flower predators evolve into these roles right away after the first appearance of large flowers?”

Authors’ response. We have overhauled the abstract into a 200-word version, following the suggestions of the Reviewers:

“Florivory, (flower consumption), occurs worldwide in modern flowering plants, associated with pollen and nectar consumption. However, florivory remains unrecorded from fossil flowers since their appearance in Early Cretaceous. We evaluated 7858 plants from seven localities of the latest Albian Dakota Formation from midcontinental North America, of which 646 specimens (8.2%) were flowers or inflorescences. Specimens were categorized into 32 morphotypes, nine which displayed 207 instances of damage from 11 insect damage types (DTs) by four functional feeding groups of hole feeding, margin feeding, surface feeding, and piercing and sucking. We assessed the same DTs inflicted by known florivores on modern flowers that also are their pollinators, and associated insect mouthpart types with the damage. The Dakota florivore and linked pollinator community was a diverse assemblage of insects. A local pattern at Braun’s Ranch showed Flower Morphotypes 4 and 5 had piercing and sucking as dominant and margin feeding as minor interactions, whereas *Dakotanthus cordiformis* at Rose Creek I and II had an opposite pattern. Although hole feeding constituted 10.1 % of all damage overwhelmingly at petal bases, bowl-shaped flowers precluded nectar robbing. These data support the early emergence of florivore and pollinator roles expressed at the local habitat and regional community levels.”

8. Editor’s comments: Introduction. The first paragraph of the introduction is fine, but the second paragraph focuses on descriptive questions that are not raised again in a clear, simple way in the conclusions. “What are” and “which” questions are not really hypothesis-testing questions, but instead purely descriptive. The authors should have a hard think about what hypotheses they are testing and how to explain them here. What are the alternatives, and how will they be distinguished?

Authors’ response. We kept the first paragraph of the Introduction intact, with the exception of inserting a definition of florivory, mentioned by one of the reviewers. As for the second paragraph, it has been recast with a yes/no hypothesis-driven theme, given the context that our manuscript is exploratory and not confirmatory in nature. The three hypothesis-driven issues—we won’t call them questions anymore—are addressed again in the Conclusions section. The recast paragraph is:

“Here, three hypotheses are posed that we seek to test in our study of Dakota Formation flowers and inflorescences. The first hypothesis is: *General features of inset florivory and related pollination, as measured by damage patterns on Dakota flowers, are very similar or the same as those made by modern florivores.* An alternative hypothesis would provide evidence for the absence of much to most of Dakota florivore damage when compared to the modern flowers. The second hypothesis is: *Major taxonomic groups of insect florivores and pollinators from the Dakota*

Formation are very similar or the same as those of today. The alternative hypothesis would posit that the taxonomic groups of florivores on Dakota flowers are substantially different than those of the modern world. The third, more focused, hypothesis is: *Nectar robbing was present on Dakota flowers.* The alternative hypothesis would provide no compelling evidence for Dakota nectar robbing insects. By addressing these three hypotheses, we place this study in a broader context of Cretaceous angiosperm pollination and provide a glimpse into early angiosperm florivory and associated pollination about 30 million years after the origin of angiosperms.”

9. Editor’s comments: Species richness. “The ‘conservative estimate’ of species richness could be backed up by a quantitative statistic. There are many ways to do this. Plant ecologists often favour Fisher’s alpha, which has advantages. I went ahead and computed alpha based on the counts in Fig. S2, which are:

292,140,53,41,36,16,13,5,5,4,4,4,3,3,2,2,2,2,1,1,1,1,1,1,1,1,1,1,1,1,1,1,1

I used the following script to compute alpha. It is based on an equation given by May (1975). `fisher<-function (n)`

```
{  
  S <- length(n)  
  N <- sum(n)  
  a <- 1  
  lasta <- 0  
  z <- 0  
  while (abs(a - lasta) > 1e-07 && z < 1000) {  
    z <- z + 1  
    lasta <- a  
    a <- S/log(1 + N/a)  
  }  
  x = N/(N + a)  
  return(a)  
}
```

I obtained 8.23, which is high but not very high based on tree inventory data in the Ecological Register: http://ecoregister.org/eco.pl?a=getPlaces&life_forms=trees&displayed_rows=25&order_by=fisher&direction=descending

An alpha of 8 is roughly indicative of a tropical dry forest, and higher than anything you would see in the temperate zone.

“There may be an error where the text mentions 645 specimens, because the above list adds up to 646.”

“The text says that Fig. S3 suggests a log normal distribution, but this is impossible to tell without logging the y-axis. Also, the log series is hard to distinguish visually from the log normal without a formal analysis. I went ahead and computed the fit of the counts to those two distributions using the `fitLogNormal` and `fitLogSeries` functions provided in the supplement to my 2015 Science Advances paper, and the data do in fact better fit the log normal. But the text should not say this without providing evidence.”

Authors’ response. We have logged the Y-axis and have replaced the figure.

10. Editor’s comments: Section 3a. “The paragraph at the end of section 3a could be cut, as Fig. S2 gives the actual counts.

Authors’ response. This paragraph has been deleted.

11. Editor's comments: Section 3b. The third and fourth paragraphs of 3b could be moved to the supplement or compacted."

Authors' response. These two paragraphs have been moved to the Electronic Supplementary Material under new Text S3.

12. Editor's comments: Section 3e. "Section 3e is very detailed and could be moved to the supplement."

Authors' response. Section 3e (Assessing functional feeding group and damage type on flower morphotypes) has been moved to the Online Supplementary Materials under Text S3.

13. Editor's comments: Section 4a. "Section 4a doesn't address a hypothesis and is descriptive. It possibly could be moved out or compacted."

Authors' response. Subsection 4a ("The Early Establishment of Angiosperm Pollination Mutualisms") has been moved to the Supplemental Data, next to table S5 (Pollination biology of 20 Cretaceous angiosperms with bowl-shaped or similar flowers), where it logically belongs. Instead we have added two paragraphs for new Subsection 4a. New Subsection 4a now addresses the two hypotheses posed in the last paragraph of the Introduction Section. The first hypothesis is "General features of insect florivory and related pollination, as measured by damage patterns on Dakota flowers, are very similar or the same as those made by modern florivores". This hypothesis is supported. The second hypothesis is: "Major taxonomic groups of insect florivores and pollinators from the Dakota Formation are very similar or the same as those of today". This hypothesis also is supported. The newly inserted Subsection 4a is:

"(a) How similar was Dakota florivory, folivores and pollinators to modern counterparts?

"The presence of four functional feeding groups subsuming 11 damage types (DTs) in Dakota Formation plants (figures 1, 2, S4, S5) provides a very modern cast to the documented florivory (table S3, text S4; figure S6). In terms of the distinctiveness of the insect-mediated damage, the mouthpart classes responsible for the damage (text S6), the proportional distribution of the damage on the flower morphotypes (Table S4), and their near identical comparison to damage produced by known modern taxa [8], affirms the hypothesis that Dakota florivory is indistinguishable from its modern equivalent. This similarity suggests a similar pollinator community, although evidence for fluid feeding, nondamaging adult taxa is circumstantial.

The richness of Dakota florivory and suggested associated pollinators provides the first extensive evidence for a community of insect visitors on the earliest, well-documented bowl-shaped flowers (table S5; figure S6; text S7). The florivore component of Dakota insect pollinators is established based on (i) distinctive DTs, (ii) extant lineages that were present during Dakota time from fossil occurrences, and (iii) evidence of relevant fossil occurrences or presence of closely related clades (phylogenetic bracketing) (texts S6, S7). The florivore assemblage consisted of a variety of insects with mandibulate and piercing-and-sucking mouthparts that produced recognizable damage patterns on flowers (text S6). This pattern affirms the hypothesis that major taxonomic groups of insect florivores were very similar to their extant counterparts (table S6; text S8). Likely associated with this community of insect damagers of floral tissue, indirect evidence indicates presence of lineages of fluid feeding, adult, and nondamager taxa that left no trace on Dakota flowers. Based on the Dakota florivory data and modern studies, the core pollinators were heteropterans (especially pentatomorphs), thrips, polyphagan beetles (principally scarab and leaf beetles, and weevils), and bees. A subordinate component of early-diverging moths, sawflies, several major lineages of nematoceran and brachyceran flies, and perhaps parasitoid wasps likely were present (table S6; text S7)."

We have replaced the extensive, historical narrative in former Subsection 4a with a more succinct, hopefully more informative overview of Cretaceous angiosperm pollination:

“(c) A brief historical perspective on Cretaceous angiosperm pollination

“The early fossil history of angiosperm pollination [2] is illustrated by bowl and similarly shaped floras by the 21 localities listed in geochronological order that provide data on inferred insect pollinator lineages and their functional-feeding-group membership (table S5, text S7). This list shows that the Dakota insect pollinator fauna likely had taxonomic similarities to that of extant basal angiosperms (table S6; texts S7, S8). The next, four-million-year-younger assemblage of pollinating insects originates from the very geographically, ecologically and taphonomically different locality of Myanmar Amber, which shows a distinct pollinator fauna (table S5, figure S7). Subsequent, Late Cretaceous floras originate from a variety of localities that reveal the expansion of dicot angiosperms and an associated pollinator fauna.”

14. Editor’s comments: Section 4b. “Section 4b does address the third question in the introduction, but the second reviewer argues that the inference is unsteady, so this section needs work.”

Authors’ response. As in changing Subsection 4a above, we similarly addressed the hypothesis that nectar feeding was present in Dakota fossils. About half for the Subsection 4b paragraph was deleted, and the other half was updated to address the issue of whether nectar robbing occurred on Dakota flowers. Our replacement paragraph is as follows:

The data on hole feeding across the flower morphotype hosts is intriguing. Hole feeding, consisting of 23 perforations of damage types DT01 and DT02 represent 10.1% of all DT occurrences among Dakota flower morphotypes. Of hole feeding occurrences, 95.2% were present on *Dakotanthus* from the Rose Creek I locality and on Flower Morphotype 4 from the Braun’s Ranch locality. There is a distinct preference for the lower half (71.4%), rather than the upper half of the petal. All holes were circular or nearly so, and no evidence of slits or tears was observed as holes on the petals. Although these data suggest that nectar robbing was present, nectar robbing would be ineffectual if flowers are open, bowl shaped, and with rewards such as nectar readily available to insect visitors [41]. All modern nectar robbing occurs with tubular or similar flowers that that are highly enclosed and have access through a narrowly throated corolla [41]. The floral morphology of Dakota flowers, exemplified by *Dakotanthus*, is inconsistent with nectar robbing, and holes are circular and rounded in contrast to modern nectar robbers that construct slits or holes with jagged outlines [42,43]. Consequently, hypothesis 3, that nectar robbing was present on Dakota flowers, is rejected. Given this outcome, a more productive search for the earliest nectar robbing would be among early occurrences of tubular or otherwise enclosed flowers in the younger Late Cretaceous [2].

15. Editor’s comments: Section 4c. “Section 4c also doesn't address a question, as currently written. It is somewhat confusing because bowl-shaped flowers dominate the authors' data set, but other kinds of flowers show up later and demonstrate pollinator associations. It's not clear why they wouldn't. So, this section needs to either be cut, compacted, or made more interesting by referencing a yes/no hypothesis.”

Authors’ response. We have moved Subsection 4c up in the Discussion Section, and it now is Section 4a. We offloaded all of the historical data in Subsection 4c to the Online Supplementary Material, where it now Text S8 (“Early Cretaceous florivores and pollinators, and their modern counterparts”). New Subsection 4a now addresses the two hypotheses posed in the last paragraph of the Introduction Section. The first hypothesis is “General features of insect florivory and related pollination, as measured by damage patterns on Dakota flowers, are very similar or the same as those made by modern florivores”. The second hypothesis is: “Major taxonomic groups of insect florivores and pollinators from the Dakota Formation are very similar or the same as those of today”.

16. Editor's comments: Conclusion. "The conclusion seems to dance around a key point, which is that the paper is exciting exactly because no similar study exists at all and because the study was made possible by the really unusual preservation of this flora. This kind of information should have been highlighted in the abstract and also towards the end of the introduction, where readers would expect to see it. So the section could possibly be moved up. A new concluding paragraph focusing on what hypothesis was tested and how a knowledge gap was filled would be useful to see."

Authors' response. We have modified the conclusion substantially, taking our cue from the newly formatted last paragraph of the Introduction and the Editor's comments immediately above. Our modified conclusion is:

"5. Conclusion

"The data on hole feeding across the flower morphotype hosts is intriguing. Hole feeding, consisting of 23 perforations of damage types DT01 and DT02 represent 10.1% of all DT occurrences among Dakota flower morphotypes. Of hole feeding occurrences, 95.2% were present on *Dakotanthus* from the Rose Creek I locality and on Flower Morphotype 4 from the Braun's Ranch locality. There is a distinct preference for the lower half (71.4%), rather than the upper half of the petal. All holes were circular or nearly so, and no evidence of slits or tears was observed as holes on the petals. Although these data suggest that nectar robbing was present, nectar robbing would be ineffectual if flowers are open, bowl shaped, and with rewards such as nectar readily available to insect visitors [41]. All modern nectar robbing occurs with tubular or similar flowers that that are highly enclosed and have access through a narrowly throated corolla [41]. The floral morphology of Dakota flowers, exemplified by *Dakotanthus*, is inconsistent with nectar robbing, and holes are circular and rounded in contrast to modern nectar robbers that construct slits or holes with jagged outlines [42,43]. Consequently, hypothesis 3, that nectar robbing was present on Dakota flowers, is rejected. Given this outcome, a more productive search for the earliest nectar robbing would be among early occurrences of tubular or otherwise enclosed flowers in the younger Late Cretaceous [2]."

Reviewer Board Member 2

17. General comments. "Comments to Author(s): The manuscript presents a data set of apparently high palaeobiological importance, and it is apparent that there are no comparable studies, the samples are highly unusual, and much has been done to document them. However, much more could be done to put the work in context, especially in the abstract, introduction, and conclusions. The stated hypotheses are also narrowly focused, and should be recast to emphasize the contrasting possibilities that extensive folivory interactions did or did not evolve at such an early stage of angiosperm evolution, which are referenced in the title."

Authors' response. We have modified the manuscript throughout and now feel that it has been placed in an appropriate context. We reached the conclusion that, with the exception of nectar robbing, that the florivory (as opposed to folivory mentioned above) interactions involving types of flower damage and their florivores/pollinators did evolve at such an early stage of angiosperm evolution.

Reviewer 1 (Angela Moles)

18. General comment 1: Overall evaluation. "Comments to Author(s): The manuscript presents a data set of apparently high palaeobiological importance, and it is apparent that there are no comparable studies, the samples are highly unusual, and much has been done to document them. However, much more could be done to put the work in context, especially in the abstract, introduction, and conclusions. The stated hypotheses are also narrowly focused, and should be recast to emphasize the contrasting possibilities that extensive folivory interactions did or did not evolve at such an early stage of angiosperm evolution, which are referenced in the title." study.

Author's response. One of our initial motivations for this study was to put to work insect damage types (DTs) to address issues in the fossil record other than herbivory on vegetative plant organs. We have modified accordingly the entire manuscript—no paragraph has been left untouched—to put it in a more focused context. We have outsourced to the Electronic Supplementary Material most of the generalized aspects of early angiosperm flower structure, florivory and pollination. We have also tightened our three questions of the second paragraph of the Introduction Section into three yes/no hypotheses, the first two of which are supported and third which is rejected. The title also has been updated to reflect the major thesis of our study.

19. General comment 2: Title, conclusion and evidence. “1) ALIGN YOUR TITLE/CONCLUSION AND YOUR EVIDENCE. Your title and conclusion suggest that you have studied pollination. What you really gathered data on was folivory. Your data provides lovely strong evidence for early association between flowers and invertebrates. I understand that there is a likely link between this and pollination, but that is not empirically established here. Thus, while I absolutely agree that you should make the link to many of these invertebrates likely being pollinators in the discussion, I think the rest of the paper (especially the title and conclusion, but also the introduction) need to be changed to accurately reflect the questions asked and the evidence presented. If you want to make conclusions about the relationship between pollinators and florivores, you need to do more to establish that link within the paper.”

Authors' response. We have implemented this shift in the structure of the manuscript. The new title, “Feeding on Early Cretaceous flowers by functionally diverse insects: implications for early angiosperm pollination”, reflects a shift in emphasis. This emphasis also is borne out by rephrasing our “three questions” into three yes/no hypotheses (Introduction Section), followed by our methodology of analyses (Materials and methods Section), followed by results that address the three hypotheses (Results Section), followed by extemporizing our findings that support or refute the three hypotheses (Discussion Section), and our conclusions regarding the hypotheses (Conclusion Section). We have also added a Subsection discussing the relationship between florivores and pollinators.

20. General comment 3. Results and introduction. “2) ALIGN YOUR RESULTS AND INTRODUCTION. Your main questions were: “First, what are the patterns of florivory and are these patterns similar to those made by modern florivores (text S1 of the electronic supplementary material)? Second, which insect visitors may have been responsible for florivory and closely related pollination that targeted Dakota flowers? Third, is there evidence of nectar robbing, or pollinator “cheating”, among the fossil flowers?”

Your results section 1) describes the flower morphotype abundance and diversity, 2) describes damage patterns, 3) assesses host specificity, 4) describes the geographic distribution of florivory, 5) steps through the different damage types on different flower morphotypes, 6) matches insect mouthparts with feeding damage (which actually addresses the second question), and 7) quantifies the amount of folivory in each damage class.”

“That is, there is a striking mismatch between what you set out to do and what you actually did. I think you need to either answer the questions you introduce (in the main text, not the SI, where most readers will not see it), OR introduce the questions you answer so that the reader understands why the information you present is interesting and important and novel. I personally would like to see the former approach (use the three questions as subheadings), but either approach would work.”

Authors' response. The first paragraph of our Introduction Section provides a brief overview of the context of the issues we address that involve early angiosperm pollination. The second paragraph of the Introduction Section sets the stage for what follows in terms of especially our Results Section and Conclusion Section. We address the similarity of our florivore patterns to those of the modern florivores in the Electronic Supplementary Material, where considerable effort is made to document, by reference

to a substantial literature, the biology of modern florivores. We coalesce all of the Electronic Supplementary Material data to make the following statement in Subsection 4a:

“Based on the Dakota florivory data and modern studies, the core pollinators were heteropterans (especially pentatomorphs), thrips, polyphagan beetles (principally scarab and leaf beetles, and weevils), and bees. A subordinate component of early-diverging moths, sawflies, several major lineages of nematoceran and brachyceran flies, and perhaps parasitoid wasps likely were present (table S6; text S7).”

Although evidence initially supported the presence of nectar robbing because of the presence of holes concentrated in the petal bases, Reviewer 2 pointed out that, the open, bowl-shaped nature of the flower would not support our hypothesis 3, and so nectar robbing is rejected.

We now have addressed the three hypotheses (formerly questions) in the Introduction Section in the Results Section, and now use the Electronic Supplementary Material to provide documentation, and have newly introduced Table 1 in the main text to provide a brief summary of the documentation in the Electronic Supplementary Material. We address the three hypotheses in Subsection 4a (the first two hypotheses about florivore damage and pollinator taxa) and in Subsection 4b (the third hypothesis about nectar robbing) of the Results Section. We took the former approach.

21. General comment 4. Length of results sections. “3) REDUCE THE LENGTH OF THE RESULTS SECTION. The results section is remarkably long, and you consistently give more detail in the results text than is necessary. The current level of detail makes it harder for your readers to get the main points. I think you could greatly improve the paper by moving details that the average reader doesn’t need (e.g. the detailed distribution of specimen types, and information that doesn’t actually address the questions you set out to answer – such as detailed information on the different damage types on each flower morphotype) to the supplementary materials.”

Authors’ response. We have moved all of this detailed background information to the Electronic Supplementary Material.

22. Minor comment 1. “Abstract, line 8 – do you actually have evidence for flies, moths and bees, or is this speculation? If the latter, it is appropriate to mention it in the discussion, but not to include it in the abstract.”

Authors’ response. We now make it clear in the text that we do not have direct evidence for those insects that do not leave damage—the nondamagers—such as fluid-feeding flies and moths. However, the situation for bees is a bit complex. While the earliest documented bee, from Myanmar Amber, is about four million years younger than the Dakota Formation, bees do possess mouthparts and leave damage on flowers. (Good modern examples of this are megachilid bees that leave notches on petal margins and apid bees that fashion holes in petals as nectar robbers.) We state that indirect or circumstantial evidence, mostly fossil occurrences and phylogenetic bracketing, indicate that these nondamager pollinator groups likely were present. The presence inferred or otherwise, of these nondamager, fluid-feeding, adult insects are documented in the Electronic Supplementary Material.

23. Minor comment 2. Your use of acronyms makes the paper unnecessarily difficult to read. Can you please write out damage type and functional feeding group in full at each use?

Authors’ response. This may be the only issue that we disagree with. The use of DTs has been in the paleoecological literature since 1999, when it was first introduced by Wilf and Labandeira (Response of plant–insect associations to Paleocene-Eocene warming, *Science*, **284**, 2153–2156), and has been used as an acronym by at least 80 research papers since then, including, recently, in the neoecological literature. The DT is the fundamental currency of studies that examine fossil herbivory, as it is a specifically defined type of damage, such as DT402 in the current study. We plan to continue to use DT as an acronym for damage type. Nevertheless, when necessary, we will preface the acronym with

“damage type”, as in “... damage type (DT) 123”, particularly at the beginning of manuscript sections. We also define, where possible, FFG as functional feeding group.

24. Minor comment 3. “Remove the (DT) information from the abstract – we do not need to know how many damage types you had in each category until the methods.”

Authors’ response. The DT information has been removed from the abstract.

25. Minor comment 4. “Page 6, 2nd line down, and elsewhere. This is another example of too much information. We don’t know what DT46 and DT402 are – and we don’t NEED to know this level of detail to understand the answers to your main questions. Put this level of information in the supplementary information, or possibly a table in the main text if you think it is really important – and keep the results text meaningful for your readers (ie use words not damage type codes).”

Authors’ response. The entire section of the two paragraphs, that includes this reference to DT46 and DT402 has been removed to the Electronic Supplemental Material. With regard to “damage type codes”, see our response to Point 23 above. The DT is how we describe explicitly defined insect (and mite) damage in the fossil record and are subsumed by the 12 functional feeding groups of hole feeding, margin feeding, skeletonization, surface feeding, oviposition, piercing and sucking, mining, galling, seed predation, borings, domatia, and pathogens. The DT is well known to paleoecologists and increasingly is used by neoecologists. For this contribution, almost all of the documentary data involving DTs are now in the Electronic Supplemental Material, including Table S2 (the raw data), Text S3 (definitions of each DT), Text S4 (modern analogs of each DT), Table S3 (flower morphotype and insect mouthparts of each DT), Text S5 (newly established DTs in this work), and Table S4 (percentage representation of DTs by flower morphotype).

26. Minor comment 5. “Page 6, 10th line down. Significant is a bit of a loaded word in science writing. Do you mean not statistically significant, or negligible?”

Authors’ response. This is a word that got away from us. We have replaced “significant” with “important”.

27. Minor comment 6. “On the 20th line down on page 5, you state “Two noticeable damage patterns, discussed below, are (1), certain flower morphotypes that were most florivorized; and (2), particular FFGs and DTs that had the greatest abundance at localities.” This statement gives us almost no information. Of course, some flower morphotypes got the most damage (that HAD to happen), and the second statement is completely without information.”

Authors’ response. We have deleted this uninformative sentence and have substituted parenthetically the sentence “(See tables S2, S3, and S4, and text S3 for additional details of DT occurrences.)”

28. Minor comment 7. “I suspect you tried to hide the length of this paper by using small font, not double-spacing the text and avoiding line numbers. However, these choices all made it harder to review. Can we have it in standard manuscript format next time please?”

Authors’ response. We have submitted the revised version of this manuscript using (i) 11-point font throughout (except for new Table 1 of the main text), (ii) double spacing throughout, and (iii) line spacing throughout.

29. Minor comment 8. “The figures are very low-resolution. Make sure the figures align with the intro and results also. Sorry to be so hard on you – I think this is a really really interesting topic and some fantastic data – I just think the paper could be a lot more effective in communicating the great things you found!”

Authors’ response. We have high-resolution versions of the figures that will replace the current low-resolution figures prior to submission.

Reviewer 2 (David Wagner)

30. General comment 1: General assessment. “The is a terrific summary of one of the most well-known fossil plant assemblages on the planet, one which the authors have been studying for decades. The details in the narrative reflect their deep knowledge of both the plants, paleoinsect fauna, and fossils. Between the text and the extensive supplementary documents, there is a wealth of data shared in this submission, which is reason enough to accept the manuscript. I must also admit that the collective breath (and depth) of the four authors is his humbling, and, more to the point, greatly exceeds what I could assess. Their approach is novel, i.e., using damage to flowers (by florivores) to demonstrate that there was a diverse and an abundant pollinator fauna in place the early Cretaceous association that they studied. In their words, “the study provides a new approach for the study of pollination in the fossil record.” The narrative is excellent, focused, appropriately detailed, and clean. The submission is wonderfully illustrated.”

Authors’ response. Thank you. We put a lot of work into morphotyping the flowers, identifying the damage, and analyzing the data.

31. General comment 2: Interpretations, use of the terms monophagy, oligophagy, and polyphagy. ‘I do have several suggestions that would improve the clarity of presentation and put some of the findings in a more understandable or broader context. I also share a couple instances where my interpretation would differ from that of the authors, and an instance where I disagree with their use of terms (i.e., their definitions of monophagous, oligophagous, and polyphagous). Without knowing the species of the fossils, these terms are misapplied.”

Author’s response. We address all of these concerns. See Point 6 above for details.

32. General comment 3: Tracking down phylogenies for insects that damage flowers. My single largest suggestion is that they make an effort to track down a time-dated phylogeny for Coleoptera, Diptera, Hymenoptera, and Lepidoptera to find out what flower-visiting insect families were present 100 mybp and do a better job of conveying what pollinators were present *that do not damage flowers*. And then include one solid paragraph that discusses such at the end of the Discussion to support their own title and thesis that “Early Cretaceous Angiosperms were Pollinated by a Functionally Diverse Insect Fauna.”

Author’s response. The timing and occurrences of these lineages in the fossil record are documented in the Electronic Supplementary Material, under Text S6, Text S7a and Text S7c, where the presence and timing of occurrence of relevant lineages are discussed that would have been present by the time the Dakota Formation was deposited during the latest Albian of the Early Cretaceous, 103 million years ago. Listed in the Electronic Supplemental Material are the 13 sources that document the earliest occurrences of relevant florivorous lineages of Orthoptera, Hemiptera, Thysanoptera, Coleoptera, Hymenoptera and Lepidoptera. These sources in the Electronic Supplementary Materia are now indicated in the main text. The relevant citation in new Section 4a of the main text is:

“The florivore component of Dakota insect pollinators is established based on (i) distinctive DTs, (ii) extant lineages that were present during Dakota time from fossil occurrences, and (iii) evidence of relevant fossil occurrences or presence of closely related clades (phylogenetic bracketing) (texts S6, S7).”

We note that the time-calibrated phylogenies that we have tracked down are often of broad scope that include both plant-associated and non-plant-associated clades. However, their phylogenetic breadth makes them no less applicable for ferreting out whether a certain clade was present during Dakota time. For example, a phylogeny of the broader beetle clade Polyphaga, or the more restrictive Phytophaga, if sufficiently well calibrated, should equally provide data as to whether leaf beetles and weevils were present by Dakota time. We also note that the use of calibrated phylogenies is supported by fossil occurrences of the taxon in question, providing an additional source of data.

33. General comment 4: Presence of lepidopteran of pollinators at 100 Ma. “The ones mentioned in the paper/discussion presently are a bit random and woefully incomplete. I am a lepidopterist so see obvious omissions from the discussion: micropterigids, adelids, choreutids, scythridids, sesiids, zygaenids, and many others. By 100 mybp most Lepidoptera superfamilies were present, so singling out Gelechiidae misses a bigger point. Maybe just suggest superfamilies with known pollinators? This effort/any listing need not be exhaustive. [Minor point: I would not equate Gelechiidae with leaf rollers (which are mostly tortricids and many gracillariids) as appears to be the case in the text from my reading. Gelechiidae are shelter formers... and few are folivores.]”

Author’s response. We address this issue extensively in new Table 1 of the main text, which summarizes the relevant data of the Electronic Supplementary Material. The issue is extensively addressed in Table S4, Texts S4 and S6, and more generally in Table S5 of the Electronic Supplementary Material. For some groups, such as the afore-mentioned Lepidoptera, we have opted to include larger taxonomic groups above the family level to avoid long lists of family level clades for which fossil occurrences are sparse, such as Lepidoptera. Also see our response to Point 6 above.

34. General comment 5: Presence of sawfly pollinators. “But this paragraph, if included, could mention sawflies, several lineages of nemtacerans (e.g., bibionids, culicids) and other flies, and likely a dozen or more families of beetles.”

Authors’ response. We now have mentioned sawfly pollinators, although we did not see damage that could be attributed to this group. We also have mentioned nematocerous and brachycerous flies that also were present, like sawflies, by Dakota time at 103 Ma. This brings up a salient point about damager taxa, florivores that access floral tissues and leave damage on the flowers, versus nondamager taxa, pollinators that access floral tissues (particularly nectar and pollen) that do not leave damage on floral tissues. We have made this distinction clear, which is now discussed in the text and new Table 1. (Also see Point 35 below.)

35. General comment 6: Dividing the list of pollinators into two groups. Related to above, I am not especially happy giving equal weight to bees and katydid in a single listing of potential pollinators, e.g., in last sentence of Discussion. I think the list of pollinators should be divided into two classes. Likely pollinators: bees, beetles, flies (not listed in Discussion sentence), moths, a few hemipteran families, and thrips; and then a list of the “also rans”: e.g., grasshoppers, crickets, katydid, whatever.”

Author’s response. The suggestion to list two groups is an excellent one. This has been done by inserting new Table 1 at the end of the main text, where we identify the florivore taxa that are damagers (direct evidence), and the fluid-feeding taxa that are nondamagers (indirect evidence) and do not belong to one of the four functional feeding groups of hole feeding, margin feeding, surface feeding, and piercing and sucking. This table represents a succinct summary of relevant data from the Electronic Supplementary Material. Included in Table 1 also is an assessment of the effectiveness of the pollinator groups, to address the katydid problem. Accordingly, we have added a “Pollinator Effectiveness” category, where we assess each pollinator group as to whether they have minimum, intermediate or maximum pollinator effectiveness. (Also see our response to Point 34 above.)

36. General comment 7: Disambiguating florivory from pollination. I would like to see the authors do more to disambiguate florivory from pollination. Their frequent juxtaposition in the manuscript suggests/implies a strong association when there need be none. Best to add a few sentences that convey that not all folivores are pollinators and not all pollinators are folivores, and that an important fraction of the floral damage treated in this manuscript may have little to do with pollination/pollinators. Caterpillars and katydids that do not move between plants and consume whole flowers are bad actors in this system. Across the Lepidoptera there are many lineages with caterpillars that feed on flowers that do not pollinate: caterpillars of Lycaenidae, Stiriinae (Noctuidae), *Eupithecia*, Heliothinae (Noctuidae), and flower-feeding cutworms (Noctuidae). (And were it me, I would downplay Orthoptera—they eat flowers

from the outside in, often leave after consuming petals, and only rarely would be moving pollen between flowers/plants.

Author's response. We have added the following section at the end of the Materials and Methods Section to clarify the distinction between insect florivores and pollinators:

“(d) Distinguishing florivores and pollinators

“Insect visitors to flowers are of two fundamental groups, florivores and pollinators [30]. Not all florivores are pollinators and not all pollinators are florivores, and the relationships between these two ecological guilds are complex [8]. Florivores typically leave damage on flowers, overwhelmingly on petals [31], often resulting in negative interactions [32]. However, some florivore interactions are neutral or even positive [31,33], as petals occasionally contain nutritive or highly scented tissues designed for consumption by florivores as pollinators [34,35]. Florivory can be a form of predation if plant embryonic tissues are destroyed before floral the opening of the flower, or if there is consumption of immature pollen, features that do not appear present in bowl-shaped Dakota flowers, as the damage is overwhelmingly on inner petal surfaces. Consequently, florivores such as Orthoptera (katydids), Hemiptera (aphids, bugs), Thysanoptera (thrips), Coleoptera (beetles), and Hymenoptera (sawflies, wasps, bees) with mandibulate, stylate, or similarly modified mouthparts [36], provide good proxy data for the broad spectrum of pollinator interactions on flowers [37] (Table 1). However, a substantial component such as most adult Diptera [38] and Lepidoptera are nondamagers, as they do not leave damage on flowers.”

37. General comment 8: Timing of florivory during flower development. “Somewhere in the text it might be good to admit that the timing of florivory is important. If the florivory happens before a flower opens and before the pollen/sexual tissue is mature, the florivory would equate to predation.”

Author's response. We have added new text in new Subsection 2d, where we state:

“Florivory can be a form of predation if plant embryonic tissues are destroyed before floral the opening of the flower, or if there is consumption of immature pollen, features that do not appear present in bowl-shaped Dakota flowers, as the damage is overwhelmingly on inner petal surfaces.”

38. General comment 9: Plant-host specificities of pollinators. “Strictly speaking, damage specificity can't reveal host specificity as defined (and used) in a voluminous ecological literature. The matter of diet breadth being inferred from damage is a stretch: a leap of faith if we don't know the relatedness of the hosts, which is unknown for this fossil assemblage. And using just three instances as your criterion is an exceeding arbitrary number if there are dozens of fossils for a given morphotaxon. What if the insect is a generalist but the ecosystem had only a single host in flower in bloom at the instant of the preservation event? Then, the florivore would appear to be a specialist when in fact it is not. Insects in the same genus or tribe might all make the same type of damage but have different hosts and be strictly monophagous, but these would appear to generalists. This is too loose for me to endorse and requires terms other than monophagy, oligophagy, and polyphagy. Perhaps add “apparent” to these words or find some other work around. I am not on board otherwise.”

Author's response. This is a good point. Given what is stated by Reviewer 2 above, while maintaining the importance of establishing the distribution of the DTs on the flower hosts, we are fixing this issue in three steps. First, we are abandoning the terms “monophagy”, “oligophagy”, and “polyphagy”, which puts the onus of these distinctions on the insects, for which we know nothing about. Rather, we are using the terms “specialized damage”, “intermediate damage”, and “generalized damage”, which puts the onus of the definition on the plants (i.e. flowers), which we do know something about. Third, because we do not know how the flowers and inflorescences are related to each other, we can use specialization terms that reference the actual distribution of damage on the flowers rather than inferring the host specialization of the insects, for which we are agnostic. As an aside, fortunately we do not have this problem in doing analogous studies of herbivory (rather than

florivory) with host plants in the same Dakota floras, of which we know their general phylogenetic relationships to each other. In the Rose Creek Flora, for example, the species of Austrobaileyaceae, Chloranthaceae, Lauraceae, Magnoliales, Rosidae, etcetera, are all described and we can accurately state which DTs are distributed as host specialized, of intermediate specificity, or host generalized patterns, given the standard definitions of host specificity. We have made these changes in the main text and in table S2 of the Electronic Supplementary Material.

39. General comment 10: Hole feeding and nectar robbing. “It is a reach and not very defensible to equate hole feeding with nectar robbing. Far more parsimonious (and ecologically common) interpretation for “hole boring” would be to gain entry into a flower when the flower was almost open but not yet. Most importantly, nectar robbing occurs with complex flower morphologies, where bees are excluded by the flower design, e.g., when nectar is hidden deep in the corolla. We have *open bowl-shaped flowers* here. Heck, just fly in and collect the pollen/nectar—why waste time boring through the petals? [FTR: Bees that are nectar thieves. generally make rough, jagged-edged holes; circular holes would like be assoc. with a beetle or a small caterpillar.]”

Author’s response. We completely agree and have rejected the hypothesis that nectar robbing was present on Dakota flowers. This is stated in Subsection 4b (“Is there evidence for nectar robbing?”) of the Discussion section:

“Although these data suggest that nectar robbing was present, nectar robbing would be ineffectual if flowers are open, bowl shaped, and with rewards such as nectar readily available to insect visitors [41]. All modern nectar robbing occurs with tubular or similar flowers that that are highly enclosed and have access through a narrowly throated corolla [41]. The floral morphology of Dakota flowers, exemplified by *Dakotanthus*, is inconsistent with nectar robbing, and holes are circular and rounded in contrast to modern nectar robbers that construct slits or holes with jagged outlines [42,43]. Consequently, hypothesis 3, that nectar robbing was present on Dakota flowers, is rejected. Given this outcome, a more productive search for the earliest nectar robbing would be among early occurrences of tubular or otherwise enclosed flowers in the younger Late Cretaceous [2].”

40. General comment 11: Adding a paragraph about nectar and pollen rewards at 100 Ma.

“Which reminds: Please add a paragraph somewhere about evidence for nectar being present in the various floral types. When do we/you first have clear evidence of nectar rewards in angiosperms? It is somewhat assumed/implicit that nectar is available in these flowers. But some early flowers may have lacked nectar and primarily supplied pollen as the principal reward. Please discuss this matter. It would be of obvious importance to the taxonomic composition of guild of pollinators that would have been present. Many of the suggested/implicated pollinators here require a nectar reward, others on pollen, and some both. Might even be worth mentioning such in the Introduction.”

Author’s response. We have added a paragraph to provide a context for the presence of floral structures that bear nectar or nectar-like fluids in the following paragraph:

“The only described and best-known flower from the Dakota Formation is *Dakotanthus cordiformis* [7]. This flower has five, crescent-shaped, nectariferous pads that occur at the base of the gynoecium, each of which is aligned with a sepal. *Dakotanthus*, the most abundant morphotype in our dataset, is a member of the Rosidae 1 clade [7] and apparently very similar to a modern taxon with a lobed nectary disc. Other Dakota flower morphotypes show poor development or apparent absence of nectaries or nectary-like structures. However, leaf taxa occurring in the same localities as the unaffiliated Dakota flowers and infructescences have been assigned to extant families within Austrobaileyales, Chloranthales, Canellales, Magnoliales, Laurales, and Rosidae 1 [15], which share a common pattern of fluid rewards for pollinating insects [22]. This pattern consists of: (i) staminoidal appendages (sterile stamens) that produce at their

base glandular secretions of nectar-like fluids, mucilage, or “viscous substances”; (ii) nectariferous glands at the base or tips of fertile stamens; (iii) stigmas that secrete nectar-like substances, usually at their tips; (iv) nectar secreting, parenchymatous tissue on the adaxial surfaces of petals or sepals; and (v) large, substantive glands at the base of stamens that would qualify as true nectaries [22]. From these observations, it is highly likely that Dakota flower morphotypes produced nectar or other secretory, nectar-like fluids that attracted insect florivores and pollinators.”

41. General comment 12: Clearing up apparent confusion between damage types, DTs and plant damage. “Minor gripe: text conflates DTs damage types with damage. It appears that DT is used for two different meanings: (1) for the different types (there are ?11 recognized) and (2) for number of damaged specimens. Find a way to disentangle and clarify. Please consider reversing primacy of your numbers for the ?11 different damage types and the arbitrary numbers you have selected to designate these. The reader shouldn’t have to flip to a table to figure out what kind of damage you are discussing. At least initially focus on a word description of the damage type and include your number in parentheses. At the very least it would make the text more interesting.”

Authors’ response. The DT numbers refer to distinct, well-defined, damage types that have been formally described in many publications, the first 150 of which are formally designated in the following publication, colloquially referred to as Version 3 of the *Damage Guide*.

Labandeira CC, Wilf P, Johnson KR, Marsh F. 2007 *Guide to insect (and other) damage types on compressed plant fossils (version 3.0 – Spring 2007)*. Washington, DC: Smithsonian Institution. (<http://paleobiology.si.edu/pdfs/insectDamageGuide3.01.pdf>)

The current, Version 4, of the *Damage Guide* is in preparation and adds three new functional feeding groups—domatia, borings and pathogens—and increases the number of damage types to 411. See Text S5 (“New damage types (DTs)”) of the Electronic Supplementary Material for a preview of the metadata attached to a newly erected damage type, DT405, from this study. Each damage type has an associated “type specimen” for that particular type of damage. We also note that the DT “number” is an intact entity that cannot be changed. In the example of DT405, this particular DT is associated with the data below, including a “type” image”, that will be included in Version 4 of the *Damage Guide*, where it is introduced in this publication and may be used in future publications if it is found to occur on flowers or other plant organs. DT405 will be included in Version 4 of the new *Damage Guide*.

(b) DT405: Margin Feeding

Short description: Delicate feeding damage at the lateral edge of a petal or leaf, forming a shallow V-shaped notch unaffected by veins, not trenched; the reaction tissue, from 0.5 to 1 mm wide and 1 to 2 mm deep.

Long description: Delicate feeding damage along the lateral edge of a foliage item typically a petal or leaf, forming a shallow V-shaped notch that is unaffected by veins and is not trenched nor extends deeply into the foliar item; the notch margin is characterized often by thick reaction tissue, from 0.5 to 1 mm wide (chordal length) and 1 to 2 mm deep; lacking vein stringers, necrotic tissue flaps and minor cuspules along the cut margin that occur in larger cusped excisions; the V-shaped notches often are present multiple times along a leaf margin. [Notch feeding].

DT photo: Figure 1C–G.

Ichnotaxonomy: No ichnotaxonomic designation currently has been specified.

Host specificity: 1.

Plant host: *Dakotanthus cordiformis*, Flower Morphotype 5 (unidentifiable angiosperms)

Locality: Rose Creek (loc. 15731); near Fairbury, Jefferson County, Nebraska, U.S.A.

Stratigraphy: Janssen Clay Member, Dakota Formation

Age: Late Albian Stage; Cretaceous Period (ca. 104 Ma)

DT geochronologic range: Late Albian–Recent.

Specimen: UF-12941

Repository: Florida Museum of Natural History, paleobotanical collections (FIMNH-PC); Gainesville, Florida, U.S.A.

Inferred herbivore: A weevil (Coleoptera: Curculionidae).

Keywords: Margin feeding; DT405; *Dakotanthus*; Rose Creek; Nebraska; U.S.A.; Dakota; Cretaceous; Albian; FIMNH-PC; Coleoptera; Curculionidae.

Comments: Some bees, such as leafcutter bees, make notches similar to those of weevils, but their notches tend to be rounded rather than V-shaped, and often larger.

Modern analog: The adult of the root weevil, *Otiorynchus ovatus* (Coleoptera: Curculionidae), on the petals of lilac, *Syringa* sp. (Oleaceae).

Modern analog photo: To be published in version 4 of the *Guide to Insect (and Other) Damage Types on Compressed Plant fossils*.

Literature: [81–83].

42. General comment 13: Edits in the text.. I have suggested a few edits in a Word version that I created from the pdf. The authors can draw from this what they want. Overall, this is an excellent manuscript, and I was happy to get a chance to have an early look. David Wagner”

Authors’ response. Thank you. We have implemented the edits.

43. Track-changes comment 1. “Commented [WD1]: Impressive , but ?not all flowers.”

Author’s response. It is unclear to us what is meant by “? not all flowers”. We assume that you mean that not all angiosperm flowers during the Early Cretaceous were pollinated by insects. This is correct in that we discuss in the Electronic Supplemental References two, well-known fossil flowers described by Peter Crane and David Dilcher that are wind pollinated. However, it would be too burdensome to convey this in the title, as it currently exists.

44. Track-changes comment 2. “Commented [WD2]: Maybe work florivory into title to be fair. That’s the focus of this paper. And, and as written now, the paper does a poor job of quantifying and describing the nature of the insect fauna that was present but does not leave visible damage to the flowers.”

Authors’ response. One of the other reviewers proposed just the opposite, opting for not using “florivory” in the title because it is too jargony. Hopefully the new title strikes a balance: “Feeding on Early Cretaceous by functionally diverse insects: Implications for early angiosperm

45. Track-changes comment 3. “Commented [WD3]: impressive.”

Authors’ response. The specimens included a few thousand on loan from the Florida Museum of Natural History to the National Museum of Natural History, as well as another few thousand at the Florida Museum of Natural History.

46. Track-changes comment 4. “Commented [WD4]: Add a qualifier word? Included. Not many hoppers are pollinators—qualify in Discussion. Same with katydids—usually feed from outside flower.”

Authors’ response. We have restructured the abstract and have deleted the reference to insect taxa, including hemipteran hoppers, in the abstract.

47. Track-changes comment 5. “Commented [WD5]: Use Oxford comma throughout.”

Authors’ response. We have checked the entire manuscript, and Oxford commas have been inserted religiously.

48. Track-changes comment 6. “Commented [WD6]: Jump from florivores to pollinators needs work. They are different and even antagonistic interactions.”

Authors’ response. We discuss in detail the difference between florivores and pollinators later in the text. In the last sentence of the abstract we use the nuanced phrase “... the early emergence of florivore and pollinator roles ...” which makes a separation, albeit nuanced, between these two ecological guilds.

49. Track-changes comment 7. “Commented [WD7]: NOTE: Folivores and florivores also can promote diversification, of course. In the latter case, by flowering at a different time a lineage could avoid florivory.”

Authors’ response. We realize that florivory could induce antagonisms, as well as mutualisms, mentioned by authors cited in the text. As a result of coping with antagonisms, plant lineages can respond by changing their timing in flowering. Accordingly, the last sentence of the first paragraph of the Introduction Section has been changed to:

“Such an evaluation could provide a better understanding of the role that mutualisms and antagonisms of angiosperms, effected by their insect pollinators, had on their joint diversification [9].”

50. Track-changes comment 8. “Commented [WD8]: This depends on what tissues were consumed and when the florivory happens relative to flower opening. Much florivory happened even before a flower opens.”

Authors’ response. This entire paragraph has been rewritten. (See our response to Point 8 above.) Regarding the relationship between florivory and pollination, we address this issue in a new Subsection 2d (“Distinguishing florivores and pollinators”) in detail. (For this addition, see our response to Point 6 above.) The timing of the flower opening (anthesis) and its role in florivory is mentioned later in the manuscript, in the same paragraph of new Subsection 2d.

51. Track-changes comment 9. “Commented [WD9]: Confusingly worded and likely not necessary as they are described below anyway.”

Authors’ response. This sentence has been deleted, as suggested by another reviewer.

52. Track-changes comment 10. “Commented [WD10]: This may help reader.”

Authors’ response. At the suggestion of another reviewer, this section has been offloaded to the Electronic Supplementary Material, where it now forms Text S3 {“Assessing functional feeding group and damage type on flower morphotypes”). I agree with the transfer. We changed the first part of this sentence to: “The most abundant morphotype, *Dakotanthus*, also displayed 115 DT occurrences ...”

53. Track-changes comment 11. “Commented [WD11]: wordy and emphasizes arbitrary numbers rather than describing the damaged itself which is more memorable and relevant.”

Authors’ response. At the suggestion of another reviewer, this section has been offloaded to the Electronic Supplementary Material, where it now forms Text S3 {“Assessing functional feeding group and damage type on flower morphotypes”). I agree with the transfer. The insertion of “i. e.” is inappropriate here because the damage from all four FFGs **are [and not exemplified by]** hole feeding, margin feeding, surface feeding, and piercing and sucking. We split the long, wordy, sentence into three sentences. The last two sentences of this string are:

“However, 66 (57.4%) of these occurrences were present as DT405. DT405 is characterized by distinctive, small, V-shaped notches present along the edges of the petals, indicative of highly stereotyped margin feeding.”

54. Track-changes comment 12. “Commented [WD12]: Literally, as written this indicates 65 kinds of damage types and not $n=65$ which I believe to be the intended meaning. The text use DTs for three different meanings which confuses: DT = damage types, number of specimens. And number of observed instances of damage (e.g., when >1 is recorded from same fossil). Find a way to disentangle and clarify.”

Authors’ response. At the suggestion of another reviewer, this section has been offloaded to the Electronic Supplementary Material, where it now forms Text S3 {“Assessing functional feeding group and damage type on flower morphotypes”). I agree with the transfer. We forgot to insert the word, occurrences, after the word “DTs”, which threw the reviewer off. To summarize, the three uses of the term, DT, is as follows:

DT types. As in the sentences, “There are eight DTs in this flora. They are two margin feeding DTs (DT12 and DT15), one piercing-and-sucking DT (DT46), three mining DTs (DT36 and DT173, and DT223) and two galling DTs (DT209 and DT397).”

DT occurrences. As in the sentence, “There are 32 occurrences of DT46 on this leaf.” In this usage, each occurrence is a single DT46 puncture mark.

DT name. As in the sentence, “This damage is skeletonization that is assigned to DT19 and not to DT192.”

We have made these distinctions clear in the earlier part of the manuscript, where we have inserted the following in Subsection 2c in the Materials and Methods section:

“This system uses the functional feeding group–damage type system in which the overarching unit of herbivory is the functional feeding group (FFG), examples of which are hole feeding, margin feeding, surface feeding and piercing and sucking for Dakota flower damage. Each FFG encompasses several or more damage types (DTs), which are the basic units of damage for fossil herbivory studies. A DT may be used in three ways. First, a DT may be used in terms of *DT richness*, referring to the kinds of DTs present; or as *DT occurrences*, as in the individual instances of damage of on a leaf; or as a *formal name*, such as DT405, which is a defined, specific mode of margin feeding damage.”

55. Track-changes comment 13. “Commented [WD13]: Do we know these flower were nectar producing?”

Authors’ response. At the suggestion of another reviewer, this section has been offloaded to the Electronic Supplementary Material, where it now forms Text S3 {“Assessing functional feeding group and damage type on flower morphotypes”). I agree with the transfer. Yes, these flowers undoubtedly produced nectar. (See our response to Point 40 above.)

56. Track-changes comment 14. “Commented [WD14]: Technically no—there are not 17 damage types and this is what the sentence says.”

Authors’ response. This should be “17 DT occurrences”, not the very different “17 DTs”. Now corrected. This section has been offloaded to the Electronic Supplementary Material, where it now forms Text S3. (See Point 55 above.)

57. Track-changes comment 15. “Commented [WD15]: Lost me here. A 75-word sentence is tough to follow.”

Authors’ response. At the suggestion of another reviewer, this section has been offloaded to the Electronic Supplementary Material, where it now forms Text S3 {“Assessing functional feeding group and damage type on flower morphotypes”). I agree with the transfer.

58. Track-changes comment 16. “Commented [WD16]: This clarification/explanation of DT 402 should have been detailed above at first mention.”

Authors' response. At the suggestion of another reviewer, this section has been offloaded to the Electronic Supplementary Material, where it now forms Text S3 {"Assessing functional feeding group and damage type on flower morphotypes"). I agree with the transfer.

59. Track-changes comment 17. "Commented [WD17]: How can damage specificity reveal host specificity? Insects in the same genus or tribe might all make the same type of damage but have different hosts and be strictly monophagous. This is really a reach if we don't know the relatedness of the hosts. And three is highly arbitrary number if there are dozens of fossils for a given morphotaxon. What if the insect is a generalist but the ecosystem had only a single host in flower in bloom at the instant of the preservation event. This is too loose for me to endorse and requires terms other than monophagy, oligophagy, and polyphagy. Trick business this."

Authors' response. We have addressed this issue and have proposed a reasonable fix, by abandoning the terms "polyphagy", "oligophagy", and "monophagy", which are centered on the insects, for which we are agnostic. Instead we have used the terms "generalized damage", "damage of intermediate specificity", and "specialized damage", which is based on the distribution of the DTs on the flower morphotypes. The term, damage, is used to reference these specificities, and is not based on the assumption that a particular DT is made by a particular insect species, but rather a particular insect species, or group of species, produces a particular DT.

The pertinent issue is that the distribution of a particular DT on the flower morphotypes is an important piece of data, as it tells us how florivores are targeting the host flowers. To the extent that one DT equals one insect species causing the damage, then damage specificity would be the same as insect specificity, thereby legitimating use of the terms "polyphagy", "oligophagy", and "monophagy". But, as Reviewer 2 states, we don't know this equivalence. Consequently, we have changed our terminology in the main text and Electronic Supplemental Material. However, I would direct attention to the following article, where this issue is tackled using modern DTs and the specific insect taxa that create those DTs:

Carvalho, M., Wilf, P., Barrios, H., Windsor, D.M., Currano, E.D., Labandeira, C.C., and Jaramillo, C.A. 2014. Insect leaf-chewing damage tracks herbivore richness in modern and ancient forests. *PLoS ONE*, 9(5), e94950.

See also our response to Points 6 and 38 above, where this issue is addressed.

60. Track-changes comment 18. "Commented [WD18]: OK, these are pretty convincing numbers. Still these is NOT monophagy, oligophagy, and polyphagy in any standard sense."

Authors' response. We have abandoned the terms of monophagy, oligophagy and polyphagy. Instead, have determined the levels of specialization based on the distribution of the DTs themselves, and not inferences regarding the levels of specialization of the putative insects creating that damage.

61. Track-changes comment 19. "Commented [WD19]: Statistical value would be expected in an ecology paper."

Authors' response. We have provided a χ^2 test for this, inserted parenthetically in the sentence.

62. Track-changes comment 20. "Commented [WD20]: *Such* ambiguous here: beetles + other mandibulate insects or both; maybe delete *such*."

Authors' response. The "such" has been deleted. This section of the manuscript has been moved to the Electronic Supplementary Material.

63. Track-changes comment 21. "Commented [WD21]: yes"

Authors' response. In the relevant literature, there has been considerable acknowledgement of beetles as prominent pollinators of early angiosperms.

64. Track-changes comment 22. “Commented [WD22]: Do we know how tall these plants were? A few micropterigids visit flowers for pollen, but usually low-growing plants. If nectar there would be others. Leafrollers are not Gelechiidae as in S6...Sawflies are not mentioned but they would have been very diverse at this time—USDA/Smithsonian check with David Smith (USDA/Smithsonian) as to the dietary habits of adult sawflies.”

Authors’ response. We have a pretty good idea as to how tall these plants were. The same authors on this manuscript—Xiao, Labandeira, Dilcher, and Ren—are conducting a parallel study on herbivory of the host plants on the Rose Creek flora. This flora is well known and represent mostly perennial, shrubby taxa up to about three or so meters tall and occurring adjacent a brackish estuary near the mid Cretaceous Mid-Continental Seaway. Larger trunks of compression or permineralized wood are conspicuously absent, and so no trees were present. The relevant passage is from page 10 of Upchurch and Dilcher (1990):

“This pattern implies that many of the leaf-bearing layers may represent leaf litter fossilized by the rapid and episodic deposition of overbank sediments. The small average leaf size of the Rose Creek flora relative to coeval assemblages from the Dakota Formation and elsewhere (Wolfe and Upchurch, 1987b) corroborates taphonomic evidence for brackish-water vegetation. In plants, saline habitats produce water stress similar to that of dry climates (Walter, 1973), which selects for small xeromorphic foliage (Walter, 1973; Givnish, 1979).

Source: Upchurch GR, Dilcher DL. 1990 Cenomanian angiosperm leaf megafossils, Dakota Formation, Rose Creek locality, Jefferson County, southeastern Nebraska. *U.S. Geol Surv Bull* **1915**, 1–55.

We have removed the mention of Gelechiidae and leaf rollers in Table S3 and have considered most of these small moth pollinator lineages as members of the Clade Eulepidoptera, now mentioned in Table 1 of the main text.

We have checked the literature, including Dave Smith’s research on sawflies, and have now included sawflies as a list of potential pollinators in several places of the main text and Electronic Supplementary Material.

65. Track-changes comment 23. “Commented [WD23]: No sure what these would be? Leafrollers are tortricids. I don’t see mention of the in supplementary documents.”

Authors’ response. We were mentioning here adult insects with mouthparts that do not leave damage on flowers. We included adult Tortricidae (leafroller moths) as one of these nondamager groups. their common name. We added “adult” to this sentence to differentiate it from speaking about larval insects. These would be included under Eulepidoptera in new Table 1 of the main text.

66. Track-changes comment 24. “Commented [WD24]: Yes, likely—safe conjecture.”

Authors’ response. In the relevant literature, there has been considerable acknowledgement of beetles as prominent pollinators of early angiosperms.

67. Track-changes comment 25. “Commented [WD25]: Not sure if this takes a hyphen; but, if so, search for other instances below and elsewhere.”

Author’s response. It probably does not take a hyphen. The term “hole feeding occurrences” would make the same sense whether or not a hyphen was present.

68. Track-changes comment 26. “Commented [WD26]: This is a reach and I am uncomfortable with it. Far more common would be entry into the flower when the flower was almost open but not yet. Bees probably make a rough hole. The circularity of the hole might reveal whether it was made by a beetle or larger bee or other mandibulate insect. I would bet on a beetle.”

Author’s response. This part of the paragraph has been deleted and has been replaced with a statement that there is not evidence for nectar robbing. In other words, our third hypothesis that nectar

robbing was present on Dakota flowers, is rejected. This correction has been made in the main text and Electronic Supplementary Material.

69. Track-changes comment 27. Commented [WD27]: We don't actually know this. And a leap of faith to suggest for fossil flora. Revise wording esp. the "more often"

Authors' response. The last half of the paragraph has been deleted, including this section. We note that hypothesis 3, that nectar robbing was present in the Dakota flora, is rejected.

70. Track-changes comment 28. "Commented [WD28]: This too is a reach especially as these are early branching angiosperm lineages...not Solanaceae. Unlikely, presumptive, and dismissive. As written this leaves the impression that nectar robbing is advantageous...probably not, of course!"

Authors' response. The last half of the paragraph has been deleted, including this section. We note that hypothesis 3, that nectar robbing was present in the Dakota flora, is rejected.

71. Track-changes comment 29. "Commented [WD29]: Think mid-Cretaceous takes a hyphen; middle Cretaceous no"

Authors' response. The term, "mid Cretaceous" does not take a hyphen, for a technical reason. Unlike most geological periods, such as the formally divided Early Triassic–Middle Triassic–Late Triassic and the Early Jurassic–Middle Jurassic–Late Jurassic, there is no formal Middle Cretaceous, just Early Cretaceous and Late Cretaceous. But to define that important ca. 30 million-year-long interval of time encompassing the later Early Cretaceous and earlier Late Cretaceous, also known as the Cretaceous Terrestrial Revolution (KTR), we use the informal "mid", with no upper-case "M".

72. Track-changes comment 30. "Commented [WD30]: I am not on board with and think these should be divided into two classes. Likely pollinators: bees, beetles, **FLIES**, moths, and thrips; and perhaps still others katydids and various Heteroptera. Somewhere restate one more time/place that many pollinators are not folivores to do a better job of disambiguating pollination from folivory. Their juxtaposition in the manuscript suggest a stronger association when there need be none. Flower eaters may not be pollinators. Flower predators can just be bad actors. Maybe this needs to be stated. Why katydids and not other grasshoppers and tree crickets? Bit players..."

Authors' response. The suggestion to list two groups is an excellent one. This has been done by inserting new Table 1 at the end of the main text, where we identify the florivore taxa that are damagers (direct evidence), and the fluid-feeding taxa that are nondamagers (indirect evidence) and do not belong to one of the four functional feeding groups of hole feeding, margin feeding, surface feeding, and piercing and sucking. Table 1 represents a succinct summary of relevant data from the Electronic Supplementary Material. Included in Table 1 also is an assessment of the effectiveness of the pollinator groups, to address the katydid problem. Accordingly, we have added a "Pollinator Effectiveness" category, where we assess each pollinator group as to whether they have minimum, intermediate or maximum pollinator effectiveness. (Also see our response to Point 34 above.)

Two minor notes. First, unlike adult fly and moth pollinators, which are nondamagers, all stages of thrips are damagers, as illustrated in much of the horticultural and agricultural literature. Second, we have not included some minor players, such as tree crickets, because we could not access documentation. As for grasshoppers (Acrididae), they were not present during the Early Cretaceous, unlike the more ancient katydids (Tettigoniidae), as acridid diversification was during the Cenozoic (Song et al. 2015).

73. Track-changes comment 31. "Commented [WD31]: Yes. Good."

Authors' response. Although the Conclusion Section has been rewritten to reflect the results of our three initial hypotheses, this statement has been retained in the Conclusion Section.

74. Track-changes comment 32. "Commented [WD32]: Yes. Nice."

Authors' response. The Conclusion Section has been rewritten to reflect the results of our three initial hypotheses, but this statement about a fuller understanding of early angiosperm pollination now in Subsection 4c of the Discussion Section.

* * * * *

I hope these changes are acceptable and will work with the Proceedings Team and my coauthors to see this research to publication.

Respectfully submitted,

Prof. Ren Dong
College of Life Sciences and
Academy for Multidisciplinary Studies
Capital Normal University
105 Xisanhuanbeilu
Beijing, 100048
Peoples Republic of China

Dr. Conrad Labandeira
Senior Research Scientist and
Curator of Fossil Arthropods
Department of Paleobiology
National Museum of Natural History
Smithsonian Institution
Washington, DC 20013-7012
United States of America

Appendix C

Capital Normal University
College of Life Sciences and Academy for Multidisciplinary Studies
105 Xisanhuanbeilu, Beijing 100048, China
4 May 2021

Associate Editor
Board Member
Proceedings Royal Society B Office

Dear Associate Editor and Board Member,

We are addressing the ‘Comments to Author’ issues that were brought up in our latest submission of the manuscript, ‘Early Cretaceous Angiosperms Were Pollinated by a Functionally Diverse Insect Fauna’, coauthored by Lifang Xiao, Conrad Labandeira, David Dilcher, and Dong Ren (RSPB-2021-0320.R1).

Comments to Author:

- Comment 1.** “RSPB-2021-0320. R1 constitutes a major revision of the text that is thorough and sincere. The revision and extraordinary cover letter address all major and minor issues that were raised. However, a few minor problems remain”.
Response. We have considerably overhauled the manuscript, and have made the needed, remaining changes, indicated below.
- Comment 2.** “There is some confusion as to who wrote the cover letter, which is signed by Dong and Labandeira but starts out by saying it is being submitted by Xiao. Regardless, the tracked changes indicate that Labandeira has done the heavy lifting. I think the author contributions statement is still accurate, though.”
Response. The cover letter was written by Labandeira – as is this one – with the approval and modification of all the authors. We agree that the author contributions statement is accurate in the allocation of work for the research behind the manuscript.
- Comment 3.** “The new title is okay, but "Florivory" is jargon and echoes "flowers" too strongly. "Damage" or maybe "Consumption" would be fair alternatives.”
Response. [Lines 1–2] We disagree. The basic methodology of this paper involves the central concept of insect damage on flowers. Insect damage on flowers is succinctly encapsulated by the single word, florivory, which from its Latin roots should be known to all, particularly biologists and paleobiologists.
- Comment 4.** “The abstract is still descriptive in tone, mentioning a knowledge gap in a general way but not suggesting a hypothesis. It also doesn't explain the unusual nature of the floral assemblage compared to everything else in the palaeobotanical record, which is arguably the main point of excitement about the paper. I earlier suggested mentioning a question such as whether "modern

pollinators and flower predators [evolved] into these roles right away after the first appearance of large flowers". This sort of thing could still be worked in."

Response. [Lines 14–37] We agree that changes are warranted. We have modified the abstract accordingly, while keeping in mind that its limit is 200 words.

5. **Comment 5.** "The new second half of the introduction is fine, but belabours the "hypothesis" language. Some trimming of the "alternative hypothesis" sentences is warranted."

Response. [Lines 63–70] We deleted the sentences referring to alternate hypotheses, which was suggested by a previous reviewer. It should be clear to a reader from each of our three stated null hypothesis as to what the alternate hypothesis is.

6. **Comment 6.** "Reviewer 2 wanted to see a time-dated phylogeny of the major holometabolous insect groups. No specific step in this direction has been taken, but the supplement now includes lengthy discussions of the roles and first appearances of insect groups, and I think this is sufficient."

Response. [See Electronic Supplementary Material] We do not know of what use is a time-dated phylogeny *among* major insect clades—presumably the reference here is to insect orders—would be toward understanding which particular phytophagous families and superfamilies were around during the late Early Cretaceous. We feel that a much more appropriate approach would be to reference which particular phytophagous lineages were present during Dakota time *within* major insect clades. An even better approach would be to document the suspect phytophagous clades in the fossil record that are recorded in deposits before or immediately after the Dakota Formation was deposited. All of this documentation is laid out in the Electronic Supplementary Material.

7. **Comment 7.** "The "conservative estimate" of species richness based on uncertain reasoning remains unchanged, despite the fact that the authors could present their own, new estimate, as suggested. Specifically, it would be a good idea to briefly mention the Fisher's alpha figure for this data set in the main body of text (it is 8.226521, as can be confirmed by using the R library *sads*). The reason is that plant ecologists will immediately grasp the meaning of this number, instead of wondering where the broad range comes from and what it might mean in a standard ecological context."

Response. [Lines 101–107] Thanks for your suggestion. We tested these specimens by the Fisher alpha function and acquired an updated value of 7.936493. The value is a bit different from the previous value of 8.226521. The reason for this difference is that the total number of specimens is 645, rather than 646. We have 35 species/morphotypes in the study, including three unidentified taxa. We focus on the 32 classified morphotypes, rather than total of 36 flower or flower reproductive organ taxa. Because too many morphotypes that have just 1 specimen, it was easy to make a mistake. The R code is referenced in the supplemental material, for which we provide a link that is accessible to reviewers.

The newly changed result in result (d) is:

"The number of florivored to total flower morphotypes at each locality (table S2) – three of ten (30%) at Rose Creek I and II, and seven of 13 (53.8%) at Braun's Ranch – are distinctly significant subsets (Fisher's alpha =7.94) of the number of available hosts at each locality."

8. **Comment 8.** "There appears to have been a mixup with respect to Fig. S3. The y-axis of this figure needs to be logged, and the response letter says it has been. However, the original figure remains in the supplemental file. This is a serious problem because the text continues to suggest that the distribution is log normal, which can't be supported based on such a representation of the data. I

do believe that the distribution does not follow the log series and is instead close to log normal, but readers should be able to see the evidence in graph form.”

Response. [Lines 169–170] We have logged the Y-axis and have replaced the figure this time in the Online Supplementary Materia.. We are agree that the distribution closer to the logarithmic normal distribution. We updated the figure in the supplementary data. On page 5, lines 77–78, we state that:

“Based on the diversity and abundance of floral morphotypes (figure S2), our estimate from a Fisher’s alpha test is 645 species (figure S3).”

The newly changed result in figure S3 is represented by the following frequency distribution. The green bars represent the frequency for each specimen, and the curved line was fitted from the logged values. Major and minor tick marks, and the fitting formula are provided.

9. **Comment 9.** “Despite the above, I congratulate the authors on a job well done.”

Response. We have endeavored to present an accurate representation of our results in this research. Please let Ren Dong at Capital Normal University or Conrad Labandeira at the National Museum of Natural History know if there are issues we need to address further.

Sincerely,

Prof. Ren Dong
 College of Life Sciences and
 Academy for Multidisciplinary Studies
 Capital Normal University
 Beijing, 100048, China

E-mail: rendong@cnu.edu.cn

Dr. Conrad C. Labandeira
Department of Paleobiology
National Museum of Natural History
Smithsonian Institution
Constitution Avenue at Tenth Street
Washington, DC, 20013, U.S.A.
E-mail: labandec@si.edu